# Complex interventions to implement a diabetic retinopathy care pathway in the public health system in Kerala: the Nayanamritham study protocol

Sobha Sivaprasad [1,2] Gopalakrishnan Netuveli,[3] Raphael Wittenberg,[4] Rajan Khobragade,[5] Rajeev Sadanandan,[5] Bipin Gopal,[6] Lakshmi Premnazir,[7] Dolores Conroy,[2] Jyotsna Srinath,[3] Radha Ramakrishnan,[2] Simon George [8] Vasudeva Iyer Sahasranamam,[8] Nayanamritham Project Collaborators

For numbered affiliations see end of article.

**Correspondence to**
Sobha Sivaprasad;
senswathi@aol.com

## ABSTRACT

**Introduction** Using a type 2 hybrid effectiveness-implementation design, we aim to pilot a diabetic retinopathy (DR) care pathway in the public health system in Kerala to understand how it can be scaled up to and sustained in the whole state.

**Methods and analysis** Currently, there is no systematic DR screening programme in Kerala. Our intervention is a teleophthalmology pathway for people with diabetes in the non-communicable disease registers in 16 family health centres. The planned implementation strategy of the pathway will be developed based on the discrete Expert Recommendations for Implementing Change taxonomy. We will use both quantitative data from a cross-sectional study and qualitative data obtained from structured interviews, surveys and group discussions with stakeholders to report the effectiveness of the DR care pathway and evaluation of the implementation strategy.

We will use logistic regression models to assess crude associations DR and sight-threatening diabetic retinopathy and fractional polynomials to account for the form of continuous covariates to predict uptake of DR screening. The primary effectiveness outcome is the proportion of patients in the non-communicable disease register with diabetes screened for DR over 12 months. Other outcomes include cost-effectiveness, safety, efficiency, patient satisfaction, timeliness and equity. The outcomes of evaluation of the implementation strategies include acceptability, feasibility, adoption, appropriateness, fidelity, penetration, costs and sustainability. Addition of more family health centres during the staggered initial phase of the programme will be considered as a sign of acceptability and feasibility. In the long term, the state-wide adoption of the DR care pathway will be considered as a successful outcome of the Nayanamritham study.

**Ethics and dissemination** The study was approved by Indian Medical Research Council (2018-0551) dated 13 March 2019. Study findings will be disseminated through scientific publications and the report will inform adoption of the DR care pathway by Kerala state in future.

**Trial registration number** ISRCTN28942696.

## Strength and limitations of this study

► This study will examine the clinical and cost effectiveness of a new diabetic retinopathy care pathway at the patient, clinician and service levels and evaluate the implementation strategy within a resource constrained environment.
► This type 2 hybrid effectiveness-implementation study will use mixed methods as method of evaluation.
► The specific actions in the implementation strategy are based on the Expert Recommendations for Implementing Change taxonomy.
► The study outlines the economic evaluation of the costs of the diabetic retinopathy care pathway.
► The study is limited by the absence of a comparator due to lack of previous data on the prevalence of diabetic retinopathy in the public health system.

## INTRODUCTION
### Background

The triad of diabetes, blindness and poverty is an urgent problem that needs an effective response in the Development Assistance Committee-listed countries as these countries are home to 80% of people with diabetes.[1] Diabetic retinopathy (DR) is one of the most common complications of diabetes.[2] Sight-threatening diabetic retinopathy (STDR) is a common cause of blindness in working-age people and unless this condition is managed early, it has considerable impact on the quality of life and productivity of the person and their family, as well as substantial financial costs to health systems.[3] In its early stages, STDR may be asymptomatic. Therefore, DR screening programmes are essential to identify STDR to enable timely treatment.[4] In DR screening programmes in high-income countries, people with diabetes are systematically screened using office-based retinal cameras

and the retinal images are graded according to severity of DR by dedicated accredited screeners and graders. Patients with STDR are identified and referred to ophthalmologists for timely treatment with laser and/or intravitreal injections of antivascular endothelial growth factor.[4] In order to successfully reduce the risk of blindness due to DR in low-middle income countries (LMICs), without a well-developed primary care infrastructure, the pathway needs to begin with systematically screening for diabetes, educating the public and healthcare professionals on early detection[5] and timely treatment of STDR and the need to optimally treat the risk factors for DR such as hyperglycaemia, hypertension and hyperlipidaemia.[2] Moreover, DR screening and treatment are challenging due to the required technology and technical expertise needed to grade retinal images and deliver costly treatment options, adding to the cost and complexity of the required interventions.

## Diabetes in Kerala

Kerala is the most advanced state in India in terms of literacy, health, social uplift and demographic transition.[6] This has been accompanied by high prevalence of diabetes. A recent report from Kerala suggests that one in five of the Kerala adult population may have diabetes.[7] The Government of Kerala launched the Aardram Mission in 2017 to transform and gear up the State's public healthcare system to achieve the sustainable development goals (SDGs) in phases with short-term goals on building infrastructure and quality care services. The overarching objectives of the Aardram Mission included providing equitable, affordable and quality care to citizens from all socioeconomic strata; strengthening the public care system by decentralising healthcare from the secondary and tertiary levels to primary care-led services and initiating preventive medicine to address the impact of non-communicable diseases, especially hypertension and diabetes. Primary care centres have been converted to family health centres (FHCs) with three doctors and four nurses in each FHC assigned to provide individual care plans to the allocated population whose records are tracked through a recently established electronic health records (EHR) called eHealth. The transformation of primary care with a focus on non-communicable diseases provided the backdrop to the implementation of a DR care pathway.

## The rationale for a complex DR care pathway

The Government of Kerala is pressing forward to achieve universal health coverage and address the SDG on poverty (SDG 1), health access (SDG 3), education (SDG 4) and gender equality (SDG 5).[8 9] Therefore, there is an urgent need to tackle the complications of diabetes. Systematic DR screening has been shown to be effective in reducing blindness.[3] However, in high-income countries, DR screening is achieved by dedicated services due to the technicalities and expertise required in DR screening and evaluation.[10–12] Introducing an isolated DR care pathway will not be sufficient to address the challenge in LMICs as these countries need to initiate holistic screening service

for diabetes and all its complications simultaneously and embed DR screening within the primary care.

## Rationale for an implementation strategy

Each aspect of the DR screening and care pathway has to be adapted to local needs and resources, requiring a locally appropriate implementation strategy. For example, mydriasis (pupil dilatation) is compulsory in some screening programmes but not in others and this process requires local approval by stakeholders.[10–12] In Kerala, mydriasis needs to be approved by the health department. Other issues faced by LMICs are that the number of undiagnosed diabetes cases is high,[13] the non-availability of a state-wide diabetes register with recall facilities, the lack of resources for standardised non-portable retinal cameras, limited capacity of staff in primary care to screen for DR and that the number of people with diabetes is larger than the capacity of the services provided by the secondary care hospitals.[14–16] Research capacity and capability are in their infancy in LMICs and implementing change in a busy environment when demands on staff are great and resources are limited is challenging.[15 16] In addition, as cataract is more prevalent in LMICs,[17] the proportion of ungradable retinal images is higher compared with developed countries.[17–19] Therefore, we need to evaluate the implementation strategy using both qualitative and quantitative methods.[20–22]

## Aims and objectives

The Nayanamritham study is facilitated by a UK-Government of Kerala partnership as a part of the ORNATE-India project funded by Global Challenges Research Fund (GCRF) and UK Research and Innovation (UKRI). We aim to introduce a DR care pathway that spans primary, secondary and tertiary care in the public health system in Kerala in a pilot study in Thiruvananthapuram. The proposed study will (1) examine the clinical and cost effectiveness of the DR care pathway at the patient, clinician and service levels and (2) evaluate the implementation strategy of the pathway.

## METHODS
### Design

We chose a type 2 hybrid effectiveness-implementation design to evaluate the effectiveness of the clinical interventions and the implementation strategies.[20–23] Mixed methods will be used as the method of evaluation. DR care pathway was developed by the Kerala Health Secretary, non-communicable diseases lead, Health and Medical Education service providers, technical and EHR teams, local authorities and the GCRF/UKRI-funded coapplicants from the UK. The DR care pathway will be set-up in a staggered approach with five FHCs initiated in the first 3 months and the remaining 11 centres will be added based on acceptance of all stakeholders and adapting and training phase for 12 months from 15 March 2018. A further 12–15 months will be allocated to recruit consenting patients to a cross-sectional study to gather quantitative data for the effectiveness outcomes. A minimal data set from this study will also be entered into

EHR to enable future screening for the consented patients. Qualitative data collection from interviews of staff, patients and focus groups at baseline and end of study and all field notes gathered during the study will be utilised to inform evaluation of the implementation strategy.[20–22]

### Target population

People with diabetes registered in the non-communicable diseases register at 16 FHCs in Thiruvananthapuram district will be invited to be screened for complications of diabetes including screening for DR. As the non-communicable diseases register is likely to show an increase in newly diagnosed diabetes as a result of training the accredited social health activists (ASHAs), in diabetes and DR, we will evaluate the effectiveness of the complex interventions delivered at each of the 16 FHCs for this study. The implementation of the pathway will take into account all patients in the non-communicable diseases register at the start of the cross-sectional study.

### Setting

Target sites for implementation of DR screening and treatment in Thiruvananthapuram will include 16 FHCs for DR screening which represents the primary care centres where patients are screened for all complications of diabetes. The retinal images, captured by the trained, resident nursing staff will be sent to a newly developed reading centre at the Regional Institute of Ophthalmology, a tertiary care centre, where newly accredited graders will grade the images. Patients with screen-positive images will be referred to three secondary care hospitals (district hospitals). Severe cases that require complex interventions will be referred to the tertiary care centre, the tertiary for specialist management of DR.

### Description of the standard of care

Currently, patients with diabetes are not systematically screened for DR in the public health system. Most patients present to the tertiary centre voluntarily either because of increased awareness of complications of diabetes or due to visual impairment. Therefore, the current standard of care will be captured as the number of patients presenting to the tertiary centre for an eye consultation for DR over a period of time as there is no baseline data in the primary and secondary care. This cross-sectional survey of the patients presenting at the FHCs will provide information on the uptake of screening of people registered in the non-communicable diseases register at the start of the study. The prevalence of DR and STDR of the screened population will be estimated from the numbers screened in all the non-communicable diseases registers during this period.

### Evidence-based clinical intervention

The new DR care pathway is the intervention and is shown in figure 1. The pathway will span primary, secondary and tertiary care. The components of the pathway are:
1. DR screening of patients with diabetes registered in the non-communicable diseases register at FHCs (primary care). The retinal images will be graded remotely at the reading centre at the tertiary centre.
2. Prompt referral for timely treatment of STDR to secondary care and tertiary centres depending on the severity of the DR.
3. Treatment of patients with sight threatening DR (at secondary and tertiary care).

### Implementation strategies

The implementation strategies of the new DR care pathway are categorised as shown in table 1 into plan, finance, education, infrastructure, quality improvement and policy contexts. These categories are developed based on the discrete Expert Recommendations for Implementing Change (ERIC) taxonomy.[20 21] The logic model is shown in figure 2.

### Outcomes

Prespecified outcomes of the effectiveness of the DR pathway and the evaluation of the implementation strategy are tabulated in tables 2 and 3, respectively.

### Data collection

#### Quantitative data

Effectiveness outcomes will originate from the data sources shown in table 2. Data collected by nurses or data operators from the EHR include age, gender, duration of diabetes, use of insulin, parental history of diabetes, other complications of diabetes including diabetic kidney disease, cardiovascular complications, and diabetic foot, random blood sugar results, urine dipstick test for albuminuria and blood pressure record. Other study-specific data collected by nurses or data operators on the day of screening include education status, occupation and income categories, and previous history of DR, cataract surgery or any other ocular history. In addition, they will measure body mass index, waist circumference and complete a lifestyle questionnaire on smoking, diet, physical activity, EQ-5D vision bolt-on.[24] The EQ-5D vision bolt-on will be used to calculate the quality adjusted life-years and utility value for economic analysis. EQ-5D alone does not capture visual acuity deficits.[25] The EQ-5D vision bolt-on asks patients to rate their health across six dimensions: mobility, self-care, usual activities, pain/discomfort, anxiety/depression and vision. Each dimension is scored in five levels: no problems, slight problems, moderate problems, severe problems and extreme problems. A recent study provided utility values based on EQ-5D vision bolt-on. The mapping was done in a clinical trial cohort with macular oedema in central retinal vein occlusion.[26]

In the reading centre, the data collection will include the grade of retinopathy in both eyes, presence of cataract and gradability of the retinal images. Data collected on referred patients will include numbers with ungradable images due to cataract, treatment options offered for DR and review appointment.

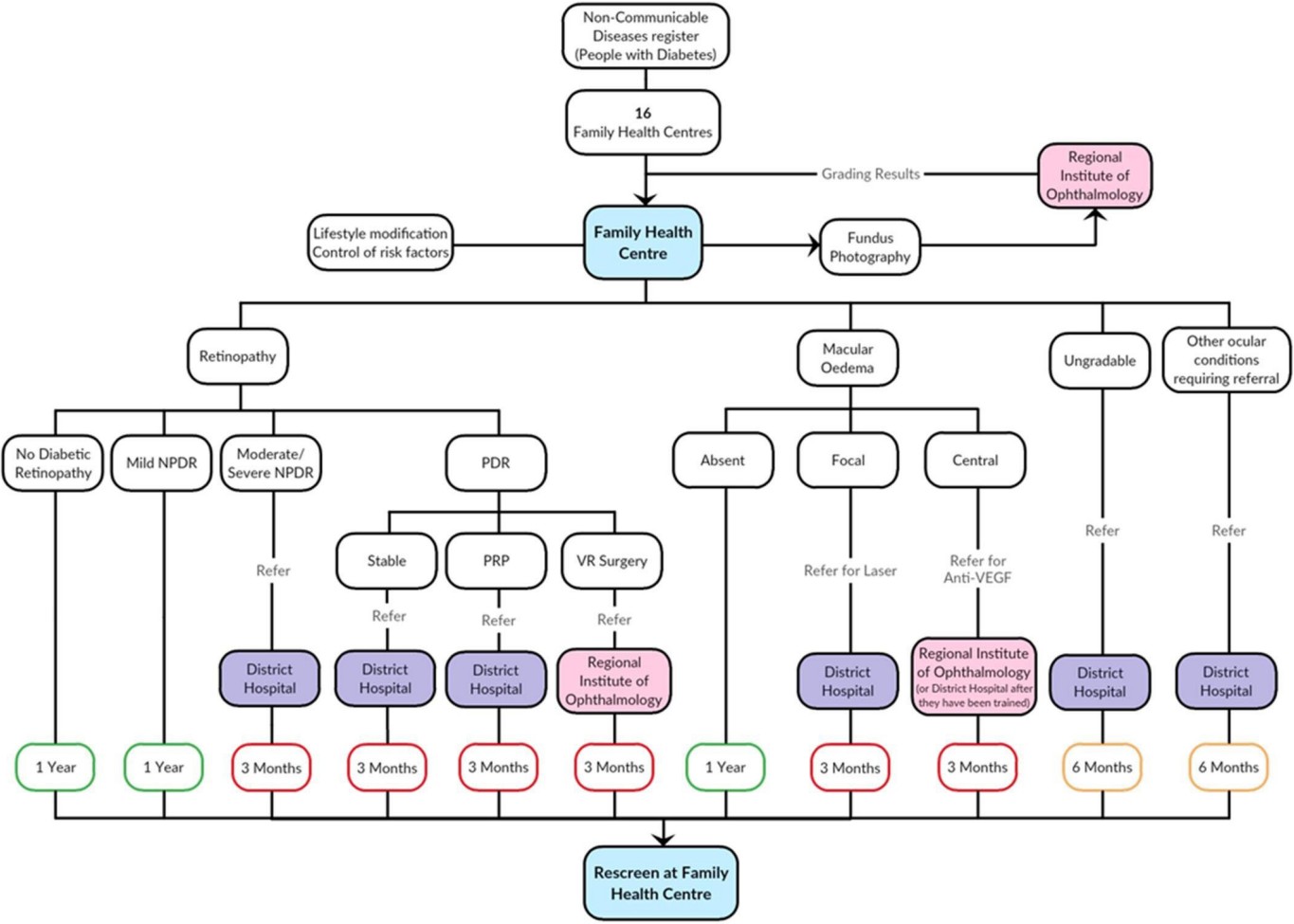

**Figure 1** Proposed diabetic retinopathy care pathway. NPDR, non-proliferative diabetic retinopathy; PDR, proliferative diabetic retinopathy; PRP, pan-retinal photocoagulation; VEGF, Vascular Endothelial Growth Factor; VR, vitreo-retinal surgery.

## Qualitative data collection and analysis

Data sources used for evaluation of the implementation strategy are shown in table 3. These will include data from a structured interview of a maximum variable sample that will reflect the context (primary, secondary and tertiary care), and the functions and skill levels of the staff (eg, nurses, doctors, etc.). Based on pragmatic considerations, at least five nurses and five primary care doctors from 16 FHCs, three to five ophthalmologists from secondary and tertiary care, two data entry operators and five ASHAs and one health service administrator should be included to get the maximum variability. Verbal consent will be obtained from these health professionals. The interviews will be conducted by an independent member from the GCRF/UKRI-funded team, who is not involved in this study, in the local language within the premises of the healthcare provider. The focus group will consist of groups of patients and staff within one FHCs. In addition, to the voice-recording of the interviews and focus group, interviewers will write field notes to describe the interview situation. The interviews and focus group content will provide the basis for the data analysis, which will be based on a descriptive phenomenological approach without data or opinion interpretation and will include transcription, condensation, coding and categorisation using qualitative analysis tools. We will use the field notes collected during the interviews to inform the understanding of the phenomenon studied.

A survey of all referred patients will be done using a structured questionnaire to evaluate their satisfaction and their perception of the barriers and facilitators. All qualitative data will be coded using NVivo and analysed using descriptive phenomenological approach following the strategy.[27] We will transcribe the interview data, identify statements or phrases, create formulated meanings or meaning units, aggregate formulated meanings and incorporate the result into descriptions.

## Data monitoring

Data will be coded before entry into the study database by the clinical teams. Only anonymised extracted data from EHR by the nurses or data entry operators will be used for analysis. Data quality will be monitored by the study project manager. Anonymised data will be checked for range checks and data quality at University of East London by the study statistics team. The ORNATE India International Advisory Board will have an overview of the conduct of the project and the Executive Group consisting of the UK-India collaborators will monitor the conduct of the study.

**Table 1** Implementation strategies

| Categories of implementation strategies[21] | Discrete implementation strategy[20] | Specific actions |
|---|---|---|
| Plan | Conduct local needs assessment | The study team will conduct a situational analysis of data from self-referred patients. |
| | Assess for readiness and identify barriers and facilitators | Assess barriers that may impede implementation of the DR care pathway, and strengths that can be used in the implementation effort by interviewing key stakeholders that is, family health centres staff, ophthalmologists in secondary and tertiary care, health service employees and patients. |
| | Develop a formal implementation blueprint | The blueprint will be used to guide the implementation of the whole DR care pathway and will be updated with time. |
| | Tailor strategies | Regular collaborative meetings will be held to tailor the implementation strategies to address barriers and leverage facilitators that are identified during various steps in the implementation. |
| | Stage implementation scale up and integration | The implementation will be staggered with five family health centres starting as pilots to enable small changes before gradually moving to other 11 centres. Addition of more centres during the staggered phase will be considered as sign of acceptability and feasibility. Ensure integration of the DR care pathway into EHR. In the long term, the adoption of the DR care pathway across the state of Kerala will be considered as success. |
| | Build a coalition | Cultivate relationships with staff at family health centres and secondary care centres, data entry operators, EHR team, project manager and the Government of Kerala health service departmental staff in the implementation effort and create a learning system to incorporate practitioner feedback. |
| | Develop academic partnerships | Partner with UK academic institutions for the purposes of shared training and bringing research skills to the implementation of the pathway. |
| | Recruit, designate and train for leadership | A project manager will be recruited to manage the programme under the supervision of the non-communicable diseases lead. The Health Secretary will conduct periodic meetings with the project team to assess performance. |
| | Obtain formal commitments | A collaborative agreement is in place between Government of Kerala and Moorfields Eye Hospital, London, UK outlining their roles and responsibilities in the implementation of the DR care pathway. |
| | Prepare patients to be active participants | Community level health workers will prepare patients to be active in their care, to ask questions and specifically to inquire about care guidelines, the evidence behind clinical decisions or about available evidence-supported treatments. Each patient will provide written consent to be screened for DR in this project as well as participate in interviews after verbal consent. |
| | Inform local opinion leaders | The Health Minister of the Kerala state and members of the local government will be briefed on the project and encouraged to be champions of the project to enable scale up to all family health centres in the state. |
| Finance | Access new funding | Use GCRF/UKRI funding to purchase smartphone retinal cameras and retinal lasers, develop EHR, train staff at family health centres and secondary care, employ project manager, data entry operators and personnel to conduct interviews, create study database linked to hand-held applications for data entry operators. State and local government funds will be sourced for scale up. |
| | Incentivisation | ASHAs will be incentivised to accompany participants with diabetes who are referred to secondary care for treatment. |
| Education | Develop educational materials | Develop DR training modules and retinal image capture modules for family health centres staff Develop laser training material to train ophthalmologists. |
| | Distribute educational materials | Distribute educational materials (including guidelines, manuals and toolkits) in person, by mail and/or electronically. Multiple training sessions are planned at each family health centres. The camera manufacturers will be invited to provide hands-on training on capturing good quality retinal images. Training on laser surgery will also be provided to the ophthalmologists at secondary care. |
| | Conduct educational meetings | Hold meetings targeting different stakeholder groups (ophthalmologists, family health centres staff, ASHAs, patients). Three educational meetings are planned by the UK team to meet with the providers in the family health centres settings to share knowledge and educate providers about the DR care pathway and its integration into their clinical practice. |
| | Achieve accreditation for graders of retinal images | Train and certify the optometrists in the Reading Centre in the tertiary centre to obtain accreditation as graders using the online international Test and Training for retinal graders. |
| | Conduct ongoing training | Nurses and doctors in the family health centres will receive on-going training on DR, pupil dilatation and capture of retinal images using low-cost smartphone retinal cameras.[29 30] |
| | Create a learning collaborative | The Health Secretary of State will foster a collaborative learning environment to improve implementation of the DR care pathway. |
| | Use train-the-trainer strategies | The UK team will first train a batch of family health centres nurses to capture retinal images and then some will be trained to train the rest of the nurses. The same cascading pathway will apply to laser training. |
| | Work with educational institutions | Moorfields Eye Hospital staff will provide the necessary guidance and share knowledge on UK diabetic retinopathy screening pathway. |
| | Make training dynamic | All staff involved will receive continual online or face-to-face training. |
| | Increase demand | The ASHAs will be trained to increase public awareness of this pathway in their house-to-house health visits. |
| | Provide ongoing consultation | The UK team from Moorfields Eye Hospital will provide ongoing consultation to support implementation of DR care pathway. |

Continued

| Categories of implementation strategies[21] | Discrete implementation strategy[20] | Specific actions |
|---|---|---|
| Infrastructure | Centralise technical assistance | The project manager and the EHR team will provide central assistance focused on implementation issues. The telemedicine project will link the patient's retinal images to EHR and transfer through a secure cloud to the tertiary centre, where the images will be graded for their DR status in a newly developed reading centre, with the results fed back to the family health centres.[31] |
| | Change physical structure and equipment | Evaluate current configuration of family health centres consulting rooms to allow suitable furniture for cameras and adapt illumination to capture retinal photographs. |
| | Change record systems | Change record systems to allow better assessment of implementation or clinical outcomes by having data entry operators interview patients and input full data into EHR within the busy family health centres clinics. |
| | Change service sites | Change the timetable for clinics in secondary care to allow a patients referred from primary care to access treatment. |
| | Revise professional roles | The non-communicable diseases lead will be delegated to take on the responsibility of implementing this DR care pathway in Thiruvananthapuram aiming for a Kerala state-wide roll out. |
| Quality improvement | Develop and implement tools for quality monitoring | Develop, test and introduce into quality-monitoring systems the right input—the appropriate language for the consent form, protocols on DR screening, grading retinal images and measures of processes, patient outcomes and implementation outcomes. |
| | Develop and organise quality monitoring systems | Develop and organise systems and procedures that monitor clinical processes and/or outcomes for the purpose of quality assurance and improvement. |
| | Audit and provide feedback | Collect and summarise clinical performance data every month and feedback to family health centres and secondary care staff to monitor, evaluate and modify pathway for acceptability to patients and staff. A continuous monitoring of retinal image quality and gradability, and the referral pathway will be done to cyclically input change into the DR pathway to improve the quality and integration. |
| | Conduct cyclical small tests of change | Staggered set up of family health centres to allow implementation of changes in a cyclical fashion using small tests of change before taking changes system-wide. Tests of change benefit will be measured continually by increase uptake of training, screening, referral and treatment. |
| | Use data experts | The EHR team will develop an application for the data entry operators to collect data. The UK collaborators will collaborate in the analysis of the data. |
| | Use data warehousing techniques | Integrate clinical records from this research project into EHR on an on-going basis. |
| | Capture and share local knowledge | Capture local knowledge from implementation sites on how staff and doctors make something work in their family health centres and then share it with other centres. |
| | Obtain and use patients feedback | The participating patients will be asked to provide feedback on the implementation effort. |
| | Promote adaptability | Identify ways the DR care pathway can be tailored to meet local needs and clarify which elements of the pathway must be maintained to preserve fidelity. |
| | Provide clinical supervision | The Director of the tertiary centre will provide the overall clinical supervision of the whole DR care pathway. He will also delegate ophthalmologists to provide ongoing supervision for the optometrists involved in grading retinal images. |
| | Provide local technical assistance | Develop and use a system to deliver technical assistance focused on implementation issues using relevant service providers. |
| | Purposely re-examine the implementation | Monthly monitoring of screened patients at each family health centre, barriers observed and small changes that are required will be implemented to continuously improve the pathway. |
| | Remind clinicians | Primary care doctors in family health centres will receive monthly newsletter on the numbers recruited across all centres to highlighting successes and providing potential solutions to issues encountered. |
| Policy contexts | Issue Government of Kerala approval to enable dilatation of pupils of patients under supervision of family health centres doctors | Ensure doctors at family health centres receive approval for mydriasis in order to deliver the DR care pathway. |
| | Mandate change | The Health Secretary of State will provide leadership to ensure the prioritisation of the implementation of the DR care pathway. |

ASHAs, accredited social health activists; DR, diabetic retinopathy; EHR, electronic health records; GCRF/UKRI, Global Challenge Research Fund/UK Research and Innovation.

## Sample size

We have chosen the proportion of patients in the non-communicable diseases register with diabetes screened for DR as our primary outcome variable.

Justification of the choice of primary outcome: this variable captures the effectiveness of our intervention as well as the fidelity of implementation. Other facts that informed our choice of the primary outcome is that we expect a short implementation time of 7–9 months during which other outcomes such as number of patients treated may not be a feasible option. Therefore, we have estimated the sample size based on this primary outcome.

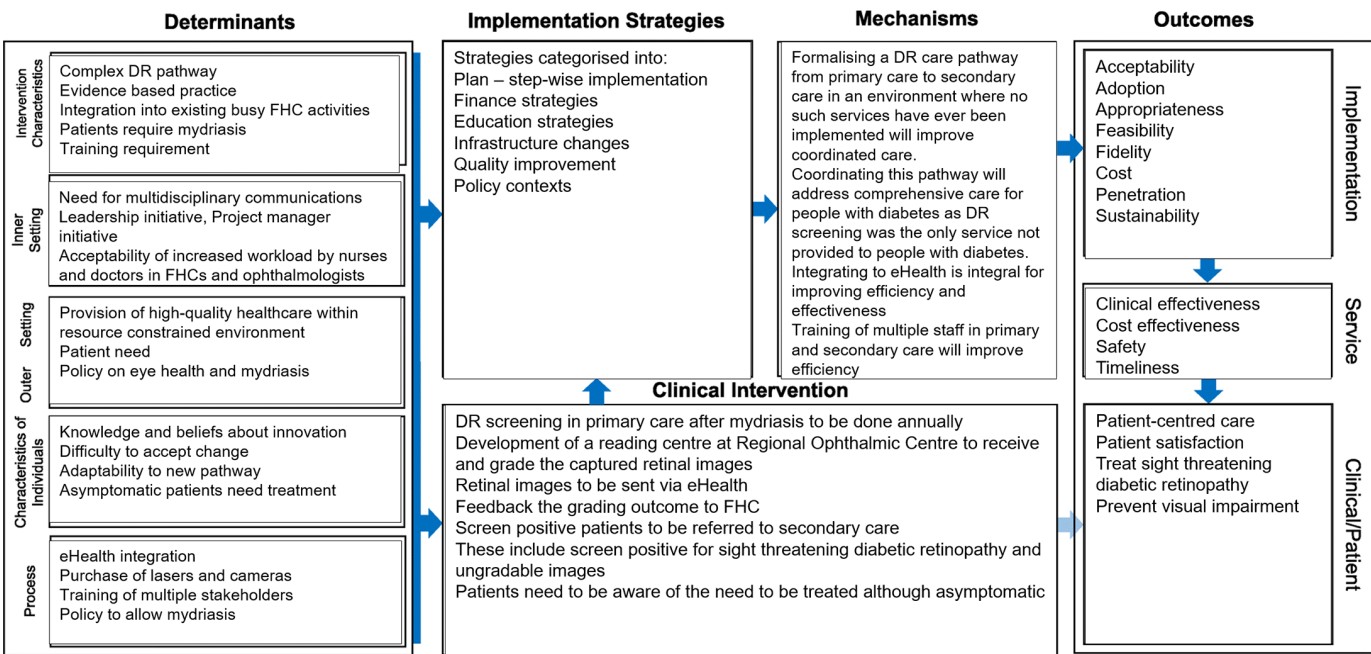

**Figure 2** The implementation research logic model. DR, diabetic retinopathy; FHC, family health centres.

Our calculations are complicated by the expected increase in number of people with diabetes in the non-communicable diseases register as public awareness of diabetes and DR increases so the denominator of numbers screened will be the number of people registered in the non-communicable diseases in each FHCs at the start of the cross-sectional study. There is no data on the baseline proportion of patients that attend the tertiary centre for screening. The prevalence of diabetes is between 10% and 16% in Thiruvananthapuram of which 8% are estimated to have DR and 3% to have STDR but about 20% may have to be referred. Assuming a finite population of 40 000 patients with diabetes, a simple random sample of 377 patients will be needed. However, we expect large design effects due to clustering patients within FHCs. There is not enough data to calculate the value of this design effect and therefore we assume it to be 3 to give a final sample size of 1131, which is equivalent to assuming a within-FHCs intracluster correlation coefficient of approximately 0.03 (n=16 FHCs) with negligible residual clustering by the intermediate cluster level of ASHAs, where mean cluster size is smaller.

As described by Becker *et al* in mammography screening,[28] we expect sources of bias in the implementation programme that will likely influence the sample size calculation. Unscreened DR cases that already exist in the community may contribute to an over-estimation of the effect of screening. We expect lead-time bias because of our short implementation period and so some DR in the community may be missed. A comparison of numbers with DR in FHCs with longer implementation period with those of shorter period may adjust for this bias. We intend to provide a descriptive analysis as well as a temporal comparison using time series analysis.

## Statistical analyses

We will create a complete case data set of the cross-sectional survey for use in the analyses of effectiveness and assess the potential impact of missing data using sensitivity analysis incorporating a range of optimistic and pessimistic scenarios for the impact of missing data. We will model two outcomes that we hypothesised would be influenced by the intervention: (1) uptake of DR screening (primary outcome) and (2) numbers of patients referred for STDR as a result of screening. We will use bivariate logistic regression models to assess crude associations between sociodemographic factors such as age, gender, education, income and living arrangements, and other clinical data with DR and STDR and use fractional polynomials to account for the form of continuous covariates to predict uptake of DR screening. Factors associated with a p<0.25 will be candidates for inclusion in the multivariable logistic regression models. We will test interactions among identified main effects to capture improvement in model prediction assessed by reduced residual variance-based statistics. To account for the staggered entry into the study, we will add a variable indicating the month of entry of each FHCs into the programme (eg, 1. if the FHC join in the first month of the programme, 2. if in the second month and so on). A non-significant coefficient for this variable will suggest the staggered approach had no effect and a significant coefficient will allow the effect to be quantified for each FHC. In our final adjusted model, we will consider associations with a p<0.05 as statistically significant. Adequacy of model discrimination and calibration will be assessed using receiver operating characteristic curves. The validation of the telemedicine will be reported as the agreement (kappa statistics) between screen positive patients graded by the graders

**Table 2** DR pathway (intervention)-related outcomes

| Outcomes | Indicators | Source of data |
|---|---|---|
| Clinical effectiveness | ► Proportion of people screened for DR in each family health centres over 12 months.<br>► Proportion of referred patients with STDR.<br>► Proportion of ungradable retinal images and percentage of cataract identified from referring these patients.<br>► Risk and complication burden in people screened for DR<br>► Presenting visual acuity of those referred from the DR pathway compared with those who self-referred to secondary care.<br>► Generalisability of the prevalence data compared with house-to-house survey in a neighbouring district. | Data from monthly entries into EHR and study database.<br>– |
| Safety | Data on complications of the DR pathway | EHR and study database |
| Cost-effectiveness | The key outcome is the number of cases of severe visual impairment or blindness due to STDR averted as a result of the new DR care pathway and resultant QALY gain.<br>Estimated QALY gain from laser treatment of STDR and from cataract surgery, using utility values from Rachapelle et al.[32]<br>Assumption that the effect of no screening is as in historic studies that reported the rate at which STDR leads to blindness if left untreated. | Data collected in the study, data from the literature and expert opinion where data could not be collected. |
| Efficiency | Efficiency is defined as optimal use of the service. For this purpose, we will first estimate the time required to process one patient from the path-process analysis. This estimate will be used to determine the number of patients who can be seen on a day. We will then calculate (1−number of patients screened)/number of patients who can be screened. Proportion can be used in a p-type control chart.[33] | Capture efficiencies in structures, resources and processes across the DR care pathway from staff interviews.<br>Path-process mining of patients screened in family health centres analysis will be based on detailed time-event logs of approximately 70 patients from the 16 FHCs |
| Patient satisfaction | Patients willingness to be screened and their satisfaction with the information provided to them and the care they received | Patient interviews and telephone survey of patients referred to secondary care for treatment |
| Timeliness | Path-process mapping at family health centres to understand delays in pathway<br>Reasons for non-attendance for those referred to secondary care for treatment of STDR | Path-process mapping at family health centres to understand delays in DR care pathway<br>Patient survey and secondary care records |
| Equity | Age and gender-based prevalence of DR and STDR will be reported on consented patients from all family health centres. Reasons of non-referral of patients that require referrals will be recorded. Numbers denied screening despite willingness to be screened will also be reported. | Data from EHR and study database. |

DR, diabetic retinopathy; EHR, electronic health records; FHCs, family health centres; QALY, quality-adjusted life year; STDR, sight-threatening diabetic retinopathy.

at the reading centre versus the DR grade as recorded by the ophthalmologists in secondary care centres. A kappa coefficient of 0.6 or higher was prespecified to indicate validity.

### Patient and public involvement

A patient and public group will be involved in plans to disseminate the study results and provide their input on the scale-up of this DR care pathway, should the implementation in Thiruvananthapuram be deemed successful by the Government of Kerala.

### ETHICS AND DISSEMINATION

The study was approved by Indian Medical Research Council (2018-0551). The study complies with the guidelines of the Declaration of Helsinki. Results of this research are expected to be disseminated through stakeholder reports and via scientific forums, specifically peer-reviewed publications and conference presentations. All participants will give written informed consent prior to entry to the study by the FHCs nurses and will be made aware that participation is strictly voluntary.

### DISCUSSION

At the conclusion of this study, we hope to assess the effectiveness and implementation outcomes of a complex DR care pathway integrating care at primary, secondary and tertiary care, covering a proportion of the diabetic population in Thiruvananthapuram. The study will set the scene for a policy for State-wide Screening and Treatment Pathway for Diabetes. Early identification of STDR and other complications due to this holistic approach and timely treatment are expected to have a positive impact on rates of blindness, chronic kidney disease, cardiovascular complications and thereby improve health, reduce multimorbidity and mortality. A DR pathway that straddles primary, secondary and tertiary levels of care, leveraging technology may have advantages of cost effectiveness and ease of implementation in LMICs compared with the current practice of detection and management of self-reported cases in tertiary centres.

This study outlines how the effectiveness of a DR care pathway and its implementation will be evaluated.[21 22] This study is timely given the increasing numbers of people with diabetes and pressure on finances available for

**Table 3** Outcomes of the evaluation of the implementation strategy

| Outcomes | Indicators | Assessment and data source |
|---|---|---|
| Acceptability | Willingness of each family health centres to be involved in the DR care pathway. Staff willingness to train patients acceptability of the pathway. | *Assessment:* Readiness to implement the DR care pathway in the 16 family health centres and any reasons for delay. *Data source:* Clinical performance from family health centres that will include monthly uptake of DR screening at each centre and any reports on barriers causing delays in implementation provided by the project manager. *Assessment:* Proportions of trained ASHAs, nurses and doctors per family health centres, increase in doctors trained in laser surgery in secondary care, proportions willing to be included in train the trainer programme. The denominator will be total numbers of each personnel available to be trained on the date of implementation. *Data sources:* Accreditation and certification records; structured interviews of at least 5 nurses and 5 primary care doctors from 16 family health centres, 3–5 ophthalmologists from secondary and tertiary care, 2 data entry operators and 5 ASHAs and 1 health service administrator to understand challenges for acceptance. *Assessment:* Screening attendances per family health centres over 12 months. *Data sources:* Clinical performance data and a telephone questionnaire survey for screened and referred patients to evaluate reasons for attendance and non-attendance. |
| Adoption | Uptake of DR care pathway at primary, secondary and tertiary care. Uptake of training by staff. Uptake of screening programme by patients. Adoption of DR care pathway by the Government of Kerala for state-wide implementation. | *Assessment:* Proportion of staff at primary, secondary and tertiary care willing to integrate DR care pathway in their workload. *Data source:* Family health centres data and staff interviews. *Assessment:* Demand training sessions; attendance rates at training sessions. *Data source:* Number of training sessions and attendance registers at training sessions. *Assessment:* Increase in proportions of patient screened over 12 months. *Data source:* EHR and study database. *Assessment:* A policy paper by the Government of Kerala for state-wide adoption of the DR care pathway as part of diabetes care for all patients. *Data source:* A detailed scale-up plan will be developed for each district based on the clinical and cost-effectiveness data from this study to inform policy. |
| Appropriateness | Appropriateness of family health centres for DR screening. Appropriateness of the complex DR care pathway across primary and secondary care. Appropriateness of referrals to secondary care. | *Assessment:* Qualitative data on barriers and facilitators of the delivery of screening and treatment in secondary care. *Data sources:* Interview of staff at family health centres and secondary care, patient telephone survey on pathway from FHCs to secondary care. *Assessment:* False positive referrals from data collection at secondary care. *Data sources:* Study database; data from patient telephone survey. |
| Cost | Costs to the Government of Kerala and societal costs of the DR screening programme and subsequent treatment. | *Assessment:* Costs of the DR screening and treatment costs in total and per person, including the costs of staff training and cameras (suitably annuitised), staff costs and travel costs of patients. *Data sources:* Data collected in the study baseline and data from other services in India and from the literature. |
| Feasibility | Screening rate per family health centres. Barriers to monthly screening uptake. | *Assessment:* Monthly increase in numbers of FHCs added to the programme. *Data source:* Interview and monthly report from project manager on facilitators and barriers on each part of the pathway. |
| Fidelity | Degree of the DR care pathway implementation. Validation of the DR grading within the telemedicine pathway. Delivery of training and the impact on the DR care pathway. | *Assessment:* Fidelity evaluation in 16 family health centres using a path-process analysis of the screening to identify the common screening pathway across centres. *Data source:* The path-process analysis will be based on detailed time-event logs of 70 patients from the 16 family health centres; a telephone follow-up of referred patients to evaluate uptake of referrals. *Assessment:* Validation of the DR grading will be assessed by the agreement of the DR grade reported by the optometrists at the reading centre in the tertiary centre and those reported by the ophthalmologists in the secondary care centres. *Data sources:* DR grading of referred patients obtained from the reading centre vs secondary care. *Assessment:* Increase in numbers of trained staff per month. *Data sources:* Staff interviews, structured observations and review of facility records. The proportions of doctors trained on laser delivery; data quality on the database; increase in non-communicable diseases registration following start of programme due to training of ASHA workers on diabetes and DR; increase in numbers of patients treated for STDR following training. |
| Penetration | Increase patient awareness of diabetes and DR and uptake of the DR screening pathway in patients with diabetes. | *Assessment:* Increase uptake of DR screening by patients registered in non-communicable diseases registers. *Data sources:* from non-communicable diseases register and EHR. |
| Sustainability | Policy makers to prepare policy on up-scale and sustainability plan for DR pathway. Integration of DR pathway in weekly activities in FHCs | *Assessment:* Availability of a policy paper from Government of Kerala on state-wide scale up of the DR care pathway. This will be considered a success of the programme. *Data sources:* Scale up plan for other districts including plan to purchase retinal cameras and acceptance of guidelines and training material for state-wide roll out. *Assessment:* Integration of DR pathway in EHR. *Data source:* EHR records of DR screening as part of non-communicable diseases care pathway. |

ASHA, accredited social health activists; DR, diabetic retinopathy; EHR, electronic health records; FHCs, family health centres; STDR, sight-threatening diabetic retinopathy.

healthcare, necessitating task shifting to enable better coverage of the population. The study will also examine the organisation functions such as structure, resources and processes that would contribute to better outcomes. We will also examine whether this additional task is feasible given the current skill levels and workload of FHCs and secondary hospitals. The study will reveal whether current health seeking behaviour of patients will support screening, especially when an invasive procedure is requested when there is no obvious impact on quality of life of the patient. By addressing a key gap in knowledge due to lack of research in this area, we will be able to decipher the barriers and facilitators that influence the successful implementation of such a programme in the public system in India. The results of this study may inform the adoption of this pathway in other areas in India and globally. However, the complexity and number of implementation strategies, local contextual factors and lack of validated implementation outcomes may limit generalisability of the results and implementation of this pathway elsewhere.

There are limitations, however, to the study design. We do not have baseline data on DR screening in the study location as these screening programmes are non-existent. For this reason, we are examining the effectiveness of the pathway in terms of presenting visual acuity for referable cases as 'proximal' outcomes. In addition, when there is no concurrent control group, causal inferences are difficult to make and may lead to both measured and unmeasured biases. When examining implementation, logistical issues, variations in healthcare facilities, socioeconomic variation and quality of healthcare personnel may likely affect our study. However, the study design avoids the ethical challenges of having a control group with no DR screening. Therefore, the study design is by necessity non-randomised and observational and will rely on newly trained staff members to collect data, which is likely to differ in completeness between the 16 FHCs. We expect an increase in referral rates after implementation of the intervention due to better public awareness, increasing knowledge of ASHAs and improved case ascertainment, at least in some FHCs . We will also examine the impact of cataract in the community and this has not been studied before in any other DR pathway. We have tried to minimise the limitations by the use of robust statistical techniques and the use of various data sources to elicit a greater understanding of how the programme will lead to better health outcomes.

One of the strengths of this study is that quantitative data backed by qualitative data will be collected to strengthen our findings and enable generalisation of our findings. Second, we will use robust statistical methods to reduce bias including selection bias and other confounders. Finally, the access to and collaboration with the UK is a key strength of the study, as it facilitates the codevelopment of the interventions from the outset. Active involvement of policy makers engaged in transformation of primary care to screen for and address complications of non-communicable diseases who value research to generate evidence for policy making and who are prepared to learn from, and adapt the DR care pathway based on implementation experience is a unique feature of this study. Findings on the mechanisms and contexts that optimise the implementation of this complex multifaceted intervention using the ERIC taxonomy will be useful to those developing and implementing these programmes in other health systems. The health economic model may highlight the health expenditure required at individual, family and Kerala State level for forecasting and planning health budgets.

Perhaps the most important aspect of the chosen evaluation is that it is built within a simultaneous developing public health strategy on population-based screening of diabetes and hypertension and a recently introduced EHR called eHealth. Conducting research in such an environment is a good example of health policy and systems research.

Despite the limitations, this study holds promise for providing high-quality data and detailed implementation information on a complex intervention in a resources-limited setting. We hope to contribute to the literature on the implementation and effectiveness of DR screening and treatment in the public health sector in LMICs.

**Author affiliations**
[1]Medical Retina Department, NIHR Moorfields Biomedical Research Centre, Moorfields Eye Hospital, London, UK
[2]Vision Sciences, UCL, London, UK
[3]Institute of Connected Communities, University of East London—Duncan House Campus, London, UK
[4]Nuffield Department of Primary Health Care Sciences, Oxford University, Oxford, UK
[5]Directorate of Health, Government of Kerala, Thiruvananthapuram, India
[6]Non-Communicable Diseases Department, Directorate of Health Services, Government Medical College Thiruvananthapuram, Thiruvananthapuram, India
[7]Directorate of Health Services, Government Medical College Thiruvananthapuram, Thiruvananthapuram, India
[8]Ophthalmology Department, Regional Institute of Ophthalmology, Government Medical College, Thiruvananthapuram, India

**Collaborators** Nayanamritham Project Collaborators: Dr Rajeevan Palpoo and Dr Chitra Raghavan will manage the referred patients in Regional Institute of Ophthalmology; Alwin Aniyankunju and Anju V Molly will be responsible for grading the retinal images. Steering Committee: Members of the International Advisory Board from Kerala.

**Contributors** All authors meet the ICJME criteria for authorship: SS, GN, RW, RK, RS, BG, LP, DC, JS, RR, SG and VIS. Conceptualisation: SS, GN, RW, RS, BG, SG and VIS; methodology: SS, GN, RR, RS, BG, VIS, SG, DC, RR and RK; acquisition of data: SS, GN, RW, DC, RR, JS, LP and BG; writing—original draft preparation: SS, DC, GN, RW, LP and RR; writing—review and editing: SS, GN, RR, RS, BG, VIS, SG, DC, RR, RK, JS and LP; funding acquisition: SS, GN, RW and RS on behalf of the collaborators.

**Funding** This work is part of the ORNATE India project funded by Global Challenges Research Fund and UK Research and Innovation through the Medical Research Council grant number MR/P027881/1.

**Disclaimer** The Director of Regional Institute of Ophthalmology contributed to study design; management, writing of the report and the decision to submit the report for publication, but have no authority over any of these activities.

**Competing interests** None declared.

**Patient and public involvement** Patients and/or the public were involved in the design, or conduct, or reporting or dissemination plans of this research. Refer to the Methods section for further details.

**Patient consent for publication**  Not required.

**Provenance and peer review**  Not commissioned; externally peer reviewed.

**ORCID iDs**

Sobha Sivaprasad http://orcid.org/0000-0001-8952-0659

Simon George http://orcid.org/0000-0002-5353-8336

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
