## [Reviewer comments · BMJ Open]

ARTICLE DETAILS

TITLE (PROVISIONAL)	Complex interventions to implement a diabetic retinopathy care pathway in the public health system in Kerala: the Nayanamritham study protocol.
AUTHORS	Sivaprasad, Sobha; Netuveli, Gopalakrishnan; Wittenberg, Raphael; Khobragade, Rajan; Sadanandan, Rajeev; Gopal, Bipin; Premnazir, Lakshmi; Conroy, Dolores; Srinath, Jyotsna; Ramakrishnan, Radha; George, Simon; Sahasranamam, Vasudeva Iyer

VERSION 1 – REVIEW

REVIEWER	Tyack, Zephaniae University of Queensland, Centre for Children's Burns and Trauma Research
REVIEW RETURNED	26-Jun-2020

GENERAL COMMENTS	The authors present a protocol for an ambitious project examining the effectiveness and implementation of a diabetic retinopathy pathway in public health settings in India. Although the project is complex, the methodology is well suited to understanding the potential effectiveness and implementation of the pathway and is generally well reported. I have a few points that could improve clarity of the manuscript and reporting: (1) From the clinical trial register it appears that this study may be nearing completion. Could the authors also add the date when the study commenced and indicate whether the study is pre-results? (2) Page 10, lines 33 – 36: “This telemedicine approach has been validated in various private or public-private ventures in India and other countries with encouraging results”. Could the validation of the approach be referenced and more detail be provided regarding the validation as this approach is central to the pathway? Are there plans to test the validation of the telemedicine approach as part of the study as this may provide data with the potential to influence more widespread adoption? (3) Page 11, line 20: “treatment outcomes of people with STDR”. Could more specific details be added regarding which outcomes referred to here or could some examples of treatment outcomes be provided? (4) Page 11, line 28-32: “...fidelity will be evaluated by adherence to the co-developed DR care pathway and efficiencies recommended in structures, resources and processes”. Could further detail be added regarding how adherence and efficiencies will be measured as having these clearly pre-specified will improve the rigour of the study?
---

	(5) Page 14: first paragraph: The author mention using field notes in relation to interviews. It could also be valuable to analyse field notes in relation to the study at each of the sites, if this was possible. (6) How will the RE-AIM framework be applied? For example, has it been used to design the process evaluation or will it be applied deductively to the qualitative data? (7) The behaviour of patients and health professionals may be critical to the success (or otherwise) of the intervention. Are there plans to use patient and health professionals to interpret and disseminate the study findings which could add to the sustainability of the pathway?
--	--

REVIEWER	Riordan, Fiona University College Cork National University of Ireland, School of Public Health
REVIEW RETURNED	13-Nov-2020

GENERAL COMMENTS	This is an interesting and ambitious pilot study which is useful formative work for the roll-out of a national screening programme. However, the paper is lacks important details on the methods and the study design. Abstract Would present following (line 29) 'The interventions will include integrating DR screening as an episode in the electronic health records system (Ehealth), multi-level training of doctors, nurses, optometrists, data operators and community health workers, increasing public awareness through health workers, provision of retinal cameras and lasers for treatment, development of protocols for the DR care pathway and establishing a retinal image grading centre at a tertiary institute' as list. Line 31 – should be 'eHealth' Line 38 – the second effectiveness outcome is unclear; will you look at visual acuity before treatment but not after? If the intervention is effective do you expect people will present with better acuity? 'Effectiveness outcomes will include numbers screened and treated in 12 months and presenting visual acuity of those who require treatment in the study period. The methods section describes the intervention and outcomes but there is no information on the methods i.e., how will you assess acceptability, feasibility, adoption etc? What data will be collected? Is the study mixed methods? Described as pilot project but what is the study design? In 'Ethics and dissemination' -you mention outputs may be adopted. As this is a pilot is there a plan to test the intervention on a larger scale? I think the aim needs to be clearer at the start of the Abstract; is it to describe the steps involved in setting up this new pathway and/or to also evaluate the pathway? Article summary Line 15 – repetition here, Kerala mentioned twice. This section is called 'strengths and limitations' but the bulleted points aren't presented as such. I think the first two point are strengths and the last a limitation, but this needs to be made clearer. 'Situational analysis' and house-to-house survey are mentioned here but not in the methods section of the Abstract. Do you mean 'Situation analysis' here?
--

	Introduction Line 12 – should be ‘one of the most common complications’ Line 15 - I think there should be a reference to support this statement Line 38 – I am not sure about this statement. It might just be the way it is phrased. I would have thought other complications (e.g. screening and treating diabetic foot) required technical expertise. Do you mean here that DR it is more complex because of the technology involved? I would suggest providing a reference to support this statement or more explanation of what is meant. Later (Line 33) this appears to be explained better. Line 17 – should be ‘eHealth’ Line 40 – should be ‘embed DR screening’ Line 41 – explain what is meant here by ‘local approval’ – the two references are from the UK Line 45 – states undiagnosed cases are high – can you provide a reference to support this statement? I would also suggest providing supporting references for the other statements made in in this sentence. The statements are made with regard to LMICs generally, so I would expect to see some examples from other LMICs cited here. Line 4 (page 6) – the point about cataract is a little long and could be clearer. Is the point that cataracts are more prevalent in LMICs than high income countries (if so, citation needed here) and therefore, more often the images taken with retinal cameras are ungradable? Line 15 – the rationale here needs to be captured in the Abstract where the aim is much less clear. From reading line 15 the purpose is to pilot a new pathway to understand how it can be scaled up and sustained? Line 14 – you present the rationale for the implementation strategy without clearly stating what the implementation strategy involves. Line 43 – design mentioned here but should also include in Abstract. You should briefly explain type 2 hybrid or provide citation e.g. https://www.sciencedirect.com/science/article/pii/S0165178119321687 If this is a type 2 hybrid you are testing both effectiveness of the clinical intervention and testing implementation strategies to support adoption of the clinical intervention. As stated in this article ‘It is important to be clear about the intervention components versus the implementation strategy components’ and ‘In hybrid type 2 designs, it is important to have an explicitly described implementation strategy that is thought to be plausible in the real world’. At this point in the paper this distinction is not made explicit. Line 53 - here the comparison is a bit clearer – this was the information missing from the Abstract on visual acuity Line 10, page 7 – this is the first mention of accredited social health activists, these need to be explained i.e., their role in the intervention. It sounds like training ASHAs is part of the implementation strategy. This relates back to earlier point about distinguishing intervention and implementation strategy. In general, I think you need a diagram or
--	--

separate section to clarify the intervention and implementation strategy. I see Figure 1 outlines the clinical pathway so perhaps just a diagram to illustrate the latter would help.

If the clinical intervention is DR screening (or the new DR screening pathway) then is the implementation strategy the 'integrating DR screening as an episode in EHRs, the multi-level training, raising public awareness, development of protocols etc' as mentioned in the Abstract? However, these are described as 'interventions' which adds to the confusion. The terminology used to refer to the clinical intervention and the strategies introduced to support implementation of this clinical intervention need to be consistent. I would suggest looking at the ERIC taxonomy to see whether you can characterise your strategies using this taxonomy <https://implementationscience.biomedcentral.com/articles/10.1186/s13012-015-0209-1>

Line 41 'Planned Diabetic Retinopathy Pathway for Implementation' outlines the clinical pathways which are made up of different clinical interventions e.g. Fundus photography, VR surgery. As such it seems as though this entire pathway is the evidence-based intervention you are trying to implement. Again, what is unclear is whether and which strategies are being used and tested to support implementation of this pathway. When you say (Line 43) 'The study will introduce pragmatic interventions to initiate a DR care pathway' which interventions are you referring to here? You need to make this clear.

Line 55 (page 7) – this section is called 'Complex interventions'. I expected a description of the interventions but instead this seems to report a mix of different things: the implementation outcomes (i.e. 'increasing numbers of people being registered') the process of implementation ('people registered will be invited for screening') and the implementation strategies ('one intervention is to train ASHAs', 'nurses and doctors will receive training'). I would suggest have clear sections where you describe and explain the following:

- Evidence-based clinical intervention / pathway (this can include the aim of the pathway and a description of how it works)
- Implementation strategies being used and tested to support this pathway (e.g. training and what else)
- Effectiveness outcomes and how they will be measured
- Implementation outcomes and how they will be measured

Line 3 (page 9) at the end of the section seems to suggest each of 1 – 6 outlined above is an intervention to be evaluated. If these are implementation strategies it would be much clearer to characterise them using the taxonomy. This would aid comparison with other studies.

Line 10 (page 9) – would suggest bullets or numbering to make the outcomes clearer. There seems to be a typo in this first sentence 'pre-specific effectiveness outcomes (included?): 1) numbers screened per FHC; 2) prevalence of DR and...'

Would start a new paragraph for implementation outcomes, perhaps again having a separate bullet for each outcome. Unlike the other outcomes, sustainability is defined but it is not clear how it will be evaluated. It might just need to revise the wording.

Line 30 (page 9) The DR pathway is described as 'co-developed' here and earlier in the Abstract, but so far in the manuscript is has not been clarified how it has been developed, how it is co-developed i.e., who has been developed with, who are the stakeholders.

Line 13 (page 10) – it should be RE-AIM not RE-ALM. I think the process evaluation and use of this framework should be mentioned earlier under study design. This is the first time process evaluation has been mentioned in the paper. If RE-AIM is guiding your evaluation, I would also suggest mentioning this in Abstract.

Line 20 (page 10) – clarify whether you plan to compare proportion on register who receive screening at start and end of the study and the duration of the study. You mention 7 to 9 months, but this is the first time the length of the study has been reported. This information should appear much earlier in the paper, under the study design.

You refer to the NCD register but state ‘paucity of information about the size of NCD register’ but earlier (line 16, page 8) you state ‘Currently, there are 51,000 people registered in the NCD register across 16 FHCs’. This is confusing and needs to be clarified. Would the FHCs not have a register from which they will identify patients.

Line 26 (page 10) – states sociodemographic and clinical data to be used in logistic regression models but it has not yet been explained where this data is from and how it is to be collected. It would make more sense to stick with the convention of reporting the section on data collection before data analysis. Also, maybe it is the way it is written but is not clear to me here what the comparator or reference group is - you are comparing a group of patients who received no intervention (referred to screening?) to those who did not – are both these groups from the 16 FHCs? This probably comes back to the fact that the study design is not clearly explained at the start of the method section and needs to be greatly improved.

Line 44 (page 11) – the data collection section is very sparse and needs work. Simply stating ‘Effectiveness outcomes will originate from the study data collected during the study period and the NCD register’ is not enough as you have not yet explained what data is recorded and available on the register, whether, what and how these data will be extracted. When you say ‘study data collated during the study period’ which data are these? Earlier you mentioned (line 36, page 7) that ‘Baseline survey of the patients presenting at the FHCs may also provide information’ but we are given no further detail on this survey. Is this the ‘house-to-house survey in a neighbouring district’ mentioned in the Article Summary section earlier (line 6 (page 3))? If so, details of this need to be included in the main text. There is no mention of the ‘situational analysis’ in the main text.

While there is a little more detail on the qualitative data collection, this is also too sparse. I expected to see a clear explanation of how each implementation outcome mentioned earlier would be assessed. I would advise including a table which clearly shows each outcome you mentioned (acceptability, appropriateness, feasibility, fidelity, and sustainability), the data used to assess each, and the method of data collection. From the brief description it seems a variety of methods will be used: a survey, observations, interviews, focus groups. You need to explain how each of these will be conducted. For example you are missing an explanation of how you will recruit and sample professionals and patients for interviews and focus groups, whether informed consent will be obtained, how you will conduct the observations (who will do this, who is to be observed). These are just some of the details required not all, and I would advise the authors to consult the relevant reporting checklists (e.g. COREQ for qualitative studies) to ensure they have covered all important details. Also, I would consider a survey to be quantitative data collection.

	When describing the implementation outcomes earlier (line 20, page 9) you mention acceptability will be measured by patient satisfaction. No further details are given so it is unclear whether this is via a survey or through interviews. Also, in this section you mention acceptability will be assessed among 'clinicians, health professionals, health department and policy makers' – it is unclear whether these are to be interviewed. Line 27 (page 9) you mention 'appropriateness and feasibility of the programme will be evaluated by the increase in service implementation and utilisation; barriers and facilitators'. Again, it is unclear how. No details are included in the data collection section. Line 9 (page 12): you describe here the approach to qualitative data analysis so it should not be under data collection. Discussion Line 38 (page 12) – typo, should read 'integrated it into primary care practice' Line 46 - this is unclear. Should it read 'this protocol outlines how the effectiveness of a DR pathway and its implementation will be evaluated'. Because of the detail lacking earlier on the study design, the nature of the implementation strategy (and distinction from the clinical intervention – DR pathway) it is difficult to rephrase. Line 50 – you need to provide supporting references for this statement. Also, I think there is a typo – should there be a full stop after 'necessitating'? 'This work is timely given the increasing numbers of people with diabetes and pressure on finances available for healthcare necessitating, task shifting will enable better coverage of the population.' Line 55 – you mention collecting data on other complications of diabetes here for the first time. This information needs to be included earlier along with a better explanation of what data will be collected and how. Line 27 – the lack of a control group is mentioned here, and you go on to state 'the study design is by necessity non-randomised and observational'. This is not a detail which should be left to the discussion section but should be stated upfront. You also mention who will collect the data, a detail which should appear earlier under data collection. Line 44 – the impact of cataract is also to be examined. Again, this is new information and should not be presented for the first here. Line 49 – triangulation of different data sources mentioned but under data analysis there is no mention of how the different data will be integrated. This goes back to the suggestion to have a table clearly showing the different outcomes, the data used to assess them, and the methods used to collect these data. This would allow the reader to see much more clearly how the different data sources feed into one another to understand how the programme is working. Line 3 (page 14) – logic models and use of theory are mentioned as strengths but again, the first time this has been mentioned. It was not made explicit that one aim is to illicit an understanding of how the intervention (or is it implementation strategy) works.
--	---

VERSION 1 – AUTHOR RESPONSE

Reviewer: 1

(1) From the clinical trial register it appears that this study may be nearing completion. Could the authors also add the date when the study commenced and indicate whether the study is pre-results? The study set-up and training of staff started on 05/09/2018 and the recruitment to the study started on 15/03/2019. The study recruitment and interviews were conducted until 15/03/2020 and then it was decided to suspend any more activities in this pilot study due to COVID-19. We are yet to complete the staff interviews for the process evaluation. There are no results as yet as the data is being collated and cleaned for statistical analysis.

(2) Page 10, lines 33 – 36: “This telemedicine approach has been validated in various private or public-private ventures in India and other countries with encouraging results”. Could the validation of the approach be referenced and more detail be provided regarding the validation as this approach is central to the pathway? Are there plans to test the validation of the telemedicine approach as part of the study as this may provide data with the potential to influence more widespread adoption? The references of previous validation are now added.
A secondary analysis of the dataset will enable us to validate this telemedicine approach. Thank you for suggesting this validation test.

(3) Page 11, line 20: “treatment outcomes of people with STDR”. Could more specific details be added regarding which outcomes referred to here or could some examples of treatment outcomes be provided?
We have added the specific treatment outcomes into the table of outcomes.

(4) Page 11, line 28-32: “...fidelity will be evaluated by adherence to the co-developed DR care pathway and efficiencies recommended in structures, resources and processes”. Could further detail be added regarding how adherence and efficiencies will be measured as having these clearly pre-specified will improve the rigour of the study?

In Kerala there is no pre-existing DR Care pathway. If such a pathway was existing we could measure adherence in terms of deviation from that pathway. In the absence of it, we will first do a path-process analysis of the screening to identify the common screening pathway used and post screening, we will do a telephonic follow-up of patients who were referred. The path-process analysis will be based on detailed time-event logs of 70 patients from the 16 FHCs.

Efficiency can be measured only on the screening part of the programme in the FHC. Rest of the care pathway, in the secondary and tertiary centres are part of the existing system. Efficiency is defined as optimal use of the service. For this purpose, we will first estimate the time required to process one patient from the path-process analysis. This estimate will be used to determine the number of patients who can be seen on a day. We will then be able to calculate $(1 - \text{number of patients screened}) / \text{number of patients that can be screened}$. proportion can be used in a p-type control chart. P-charts are usually used to study efficiency in service utilisation like surgical room occupancy.

(5) Page 14: first paragraph: The author mention using field notes in relation to interviews. It could also be valuable to analyse field notes in relation to the study at each of the sites, if this was possible. It is our intention to analyse the field notes.

(6) How will the RE-AIM framework be applied? For example, has it been used to design the process evaluation or will it be applied deductively to the qualitative data?

RE-AIM framework was used to inform the whole of the evaluation. In addition to using structured interviews and copious field notes from the project manager, ASHA workers, nurses and doctors in the FHC, ophthalmologists in the secondary and tertiary care, administrators and retinopathy graders and policy makers, we will use the collected data to test the performance of the five domains. For example, socio-demographic analysis of the patients screened will inform us about Reach. A telephone survey of all patients who were referred for further treatment was also done. Focus groups were planned but it will not go ahead due to COVID-19.

(7) The behaviour of patients and health professionals may be critical to the success (or otherwise) of the intervention. Are there plans to use patient and health professionals to interpret and disseminate the study findings which could add to the sustainability of the pathway?

We have no plans to include patients but our plan includes interpretation of [the behaviour/views?] of healthcare professionals including their solutions to the barriers.

Reviewer: 2

Comments to the Author

This is an interesting and ambitious pilot study which is useful formative work for the roll-out of a national screening programme. However, the paper lacks important details on the methods and the study design.

Abstract

Would present following (line 29) 'The interventions will include integrating DR screening as an episode in the electronic health records system (Ehealth), multi-level training of doctors, nurses, optometrists, data operators and community health workers, increasing public awareness through health workers, provision of retinal cameras and lasers for treatment, development of protocols for the DR care pathway and establishing a retinal image grading centre at a tertiary institute' as list.

Thank you. We have enumerated them now.

Line 31 – should be 'eHealth'

Corrected

Line 38 – the second effectiveness outcome is unclear; will you look at visual acuity before treatment but not after? If the intervention is effective do you expect people will present with better acuity? 'Effectiveness outcomes will include numbers screened and treated in 12 months and presenting visual acuity of those who require treatment in the study period.

The methods section describes the intervention and outcomes but there is no information on the methods i.e., how will you assess acceptability, feasibility, adoption etc? What data will be collected? Is the study mixed methods? Described as pilot project but what is the study design?

It is a mixed methods study where both quantitative data across all 16 FHCs will be collected and qualitative aspects include testing the implementation strategy and a process evaluation. These have now been added to the abstract.

Acceptability, feasibility, and adoption within the pilot project are tested by a staggered start of the screening program in the FHCs. We start the program with fewer number of FHCs and if more FHCs are added on to the programme it is considered as sign of acceptability and feasibility in the short term. In the long term, the adoption of the programme state-wide will be considered as success.

In 'Ethics and dissemination' -you mention outputs may be adopted. As this is a pilot is there a plan to test the intervention on a larger scale? I think the aim needs to be clearer at the start of the Abstract;

is it to describe the steps involved in setting up this new pathway and/or to also evaluate the pathway?

We have made it clear that the outputs will describe both the set-up and evaluation of the DR pathway.

Article summary

Line 15 – repetition here, Kerala mentioned twice.

Corrected.

This section is called 'strengths and limitations' but the bulleted points aren't presented as such. I think the first two points are strengths and the last a limitation, but this needs to be made clearer. These points are now made clearer under separate headings.

'Situational analysis' and house-to-house survey are mentioned here but not in the methods section of the Abstract. Do you mean 'Situation analysis' here?

Yes, corrected.

Introduction

Line 12 – should be 'one of the most common complications'

Corrected

Line 15 - I think there should be a reference to support this statement

Reference is now added.

Line 38 – I am not sure about this statement. It might just be the way it is phrased. I would have thought other complications (e.g. screening and treating diabetic foot) required technical expertise. Do you mean here that DR it is more complex because of the technology involved? I would suggest providing a reference to support this statement or more explanation of what is meant. Later (Line 33) this appears to be explained better.

The sentence has been edited to explain the complexity of the interventions.

Line 17 – should be 'eHealth'

Corrected

Line 40 – should be 'embed DR screening'

Corrected

Line 41 – explain what is meant here by 'local approval' – the two references are from the UK

In the UK, the commissioners need to approve mydriasis and in Kerala, mydriasis has to be approved by the health department. This has been added.

Line 45 – states undiagnosed cases are high – can you provide a reference to support this statement? I would also suggest providing supporting references for the other statements made in in this sentence. The statements are made with regard to LMICs generally, so I would expect to see some examples from other LMICs cited here.

We have added the references here.

Line 4 (page 6) – the point about cataract is a little long and could be clearer. Is the point that cataracts are more prevalent in LMICs than high income countries (if so, citation needed here) and therefore, more often the images taken with retinal cameras are ungradable?

We have edited this sentence and added a reference.

Line 15 – the rationale here needs to be captured in the Abstract where the aim is much less clear. From reading line 15 the purpose is to pilot a new pathway to understand how it can be scaled up and sustained?

I have corrected the purpose in the abstract.

Line 14 – you present the rationale for the implementation strategy without clearly stating what the implementation strategy involves.

We have added the implementation strategy

Line 43 – design mentioned here but should also include in Abstract. You should briefly explain type 2 hybrid or provide citation e.g. <https://www.sciencedirect.com/science/article/pii/S0165178119321687>

If this is a type 2 hybrid you are testing both effectiveness of the clinical intervention and testing implementation strategies to support adoption of the clinical intervention. As stated in this article ‘It is important to be clear about the intervention components versus the implementation strategy components’ and ‘In hybrid type 2 designs, it is important to have an explicitly described implementation strategy that is thought to be plausible in the real world’. At this point in the paper this distinction is not made explicit.

We have re-written the design section and included this reference. Thank you.

Line 53 - here the comparison is a bit clearer – this was the information missing from the Abstract on visual acuity

We have added it now.

Line 10, page 7 – this is the first mention of accredited social health activists, these need to be explained i.e., their role in the intervention. It sounds like training ASHAs is part of the implementation strategy.

This has now been moved up and added it to the implementation strategy.

This relates back to earlier point about distinguishing intervention and implementation strategy. In general, I think you need a diagram or separate section to clarify the intervention and implementation strategy. I see Figure 1 outlines the clinical pathway so perhaps just a diagram to illustrate the latter would help.

We have now added the tables to distinguish the tests and outcome of implementation strategy and process evaluation

If the clinical intervention is DR screening (or the new DR screening pathway) then is the implementation strategy the ‘integrating DR screening as an episode in EHRs, the multi-level training, raising public awareness, development of protocols etc’ as mentioned in the Abstract? However, these are described as ‘interventions’ which adds to the confusion. The terminology used to refer to the clinical intervention and the strategies introduced to support implementation of this clinical intervention need to be consistent. I would suggest looking at the ERIC taxonomy to see whether you can characterise your strategies using this taxonomy <https://implementationscience.biomedcentral.com/articles/10.1186/s13012-015-0209-1>

Thank you. We have now made it clearer with the help of this taxonomy.

Line 41 ‘Planned Diabetic Retinopathy Pathway for Implementation’ outlines the clinical pathways which are made up of different clinical interventions e.g. Fundus photography, VR surgery. As such it seems as though this entire pathway is the evidence-based intervention you are trying to implement. Again, what is unclear is whether and which strategies are being used and tested to support implementation of this pathway. When you say (Line 43) ‘The study will introduce pragmatic

interventions to initiate a DR care pathway' which interventions are you referring to here? You need to make this clear.

We have deleted this sentence as it is a repetition now.

Line 55 (page 7) – this section is called 'Complex interventions'. I expected a description of the interventions but instead this seems to report a mix of different things: the implementation outcomes (i.e. 'increasing numbers of people being registered') the process of implementation ('people registered will be invited for screening') and the implementation strategies ('one intervention is to train ASHAs', 'nurses and doctors will receive training'). I would suggest have clear sections where you describe and explain the following:

- Evidence-based clinical intervention / pathway (this can include the aim of the pathway and a description of how it works)
- Implementation strategies being used and tested to support this pathway (e.g. training and what else)
- Effectiveness outcomes and how they will be measured
- Implementation outcomes and how they will be measured

We have now re-framed the protocol to include the first two headers and all outcomes are now under separate heading in Table 1.

Line 3 (page 9) at the end of the section seems to suggest each of 1 – 6 outlined above is an intervention to be evaluated. If these are implementation strategies it would be much clearer to characterise them using the taxonomy. This would aid comparison with other studies.

These have now been deleted and added to the table as outcomes.

Line 10 (page 9) – would suggest bullets or numbering to make the outcomes clearer. There seems to be a typo in this first sentence 'pre-specific effectiveness outcomes (included?): 1) numbers screened per FHC; 2) prevalence of DR and...'

Would start a new paragraph for implementation outcomes, perhaps again having a separate bullet for each outcome. Unlike the other outcomes, sustainability is defined but it is not clear how it will be evaluated. It might just need to revise the wording.

In table 1, we have now added the sustainability tests and outcomes.

Line 30 (page 9) The DR pathway is described as 'co-developed' here and earlier in the Abstract, but so far in the manuscript it has not been clarified how it has been developed, how it is co-developed i.e., who has been developed with, who are the stakeholders.

This has now been clarified in the study design section.

Line 13 (page 10) – it should be RE-AIM not RE-ALM. I think the process evaluation and use of this framework should be mentioned earlier under study design. This is the first time process evaluation has been mentioned in the paper. If RE-AIM is guiding your evaluation, I would also suggest mentioning this in Abstract.

Thank you and we have added this in the abstract.

Line 20 (page 10) – clarify whether you plan to compare proportion on register who receive screening at start and end of the study and the duration of the study. You mention 7 to 9 months, but this is the first time the length of the study has been reported. This information should appear much earlier in the paper, under the study design.

I have added the exact dates to the study design section.

You refer to the NCD register but state 'paucity of information about the size of NCD register' but earlier (line 16, page 8) you state 'Currently, there are 51,000 people registered in the NCD register

across 16 FHCs'. This is confusing and needs to be clarified. Would the FHCs not have a register from which they will identify patients.

All FHCs have NCD registers and we have clarified it.

Line 26 (page 10) – states sociodemographic and clinical data to be used in logistic regression models but it has not yet been explained where this data is from and how it is to be collected. It would make more sense to stick with the convention of reporting the section on data collection before data analysis. Also, maybe it is the way it is written but is not clear to me here what the comparator or reference group is - you are comparing a group of patients who received no intervention (referred to screening?) to those who did not – are both these groups from the 16 FHCs? This probably comes back to the fact that the study design is not clearly explained at the start of the method section and needs to be greatly improved.

We have now added all the data collected.

Line 44 (page 11) – the data collection section is very sparse and needs work. Simply stating 'Effectiveness outcomes will originate from the study data collected during the study period and the NCD register' is not enough as you have not yet explained what data is recorded and available on the register, whether, what and how these data will be extracted. When you say 'study data collated during the study period' which data are these? Earlier you mentioned (line 36, page 7) that 'Baseline survey of the patients presenting at the FHCs may also provide information' but we are given no further detail on this survey. Is this the 'house-to-house survey in a neighbouring district' mentioned in the Article Summary section earlier (line 6 (page 3))? If so, details of this need to be included in the main text. There is no mention of the 'situational analysis' in the main text.

I have now added it to the data collection, the effectiveness outcome and in the methods.

While there is a little more detail on the qualitative data collection, this is also too sparse. I expected to see a clear explanation of how each implementation outcome mentioned earlier would be assessed. I would advise including a table which clearly shows each outcome you mentioned (acceptability, appropriateness, feasibility, fidelity, and sustainability), the data used to assess each, and the method of data collection. From the brief description it seems a variety of methods will be used: a survey, observations, interviews, focus groups. You need to explain how each of these will be conducted. For example you are missing an explanation of how you will recruit and sample professionals and patients for interviews and focus groups, whether informed consent will be obtained, how you will conduct the observations (who will do this, who is to be observed). These are just some of the details required not all, and I would advise the authors to consult the relevant reporting checklists (e.g. COREQ for qualitative studies) to ensure they have covered all important details. Also, I would consider a survey to be quantitative data collection.

Thank you and we have now used to this checklist to expand the qualitative data collection and analysis section.

When describing the implementation outcomes earlier (line 20, page 9) you mention acceptability will be measured by patient satisfaction. No further details are given so it is unclear whether this is via a survey or through interviews.

This survey is now included in the outcome on acceptability

Also, in this section you mention acceptability will be assessed among 'clinicians, health professionals, health department and policy makers' – it is unclear whether these are to be interviewed.

The interviews have now been described in the implementation strategy test and outcome

Line 27 (page 9) you mention 'appropriateness and feasibility of the programme will be evaluated by the increase in service implementation and utilisation; barriers and facilitators'. Again, it is unclear how. No details are included in the data collection section.

The interviews have now been described in the implementation strategy test and outcome

Line 9 (page 12): you describe here the approach to qualitative data analysis so it should not be under data collection.

This section is now modified to 'qualitative data collection and analysis'.

Discussion

Line 38 (page 12) – typo, should read 'integrated it into primary care practice'

Corrected

Line 46 - this is unclear. Should it read 'this protocol outlines how the effectiveness of a DR pathway and its implementation will be evaluated'. Because of the detail lacking earlier on the study design, the nature of the implementation strategy (and distinction from the clinical intervention – DR pathway) it is difficult to rephrase.

We have now re-phrased this sentence and added process evaluation.

Line 50 – you need to provide supporting references for this statement. Also, I think there is a typo – should there be a full stop after 'necessitating'? 'This work is timely given the increasing numbers of people with diabetes and pressure on finances available for healthcare necessitating, task shifting will enable better coverage of the population.'

We have edited this sentence.

Line 55 – you mention collecting data on other complications of diabetes here for the first time. This information needs to be included earlier along with a better explanation of what data will be collected and how.

We have included this under data collection

Line 27 – the lack of a control group is mentioned here, and you go on to state 'the study design is by necessity non-randomised and observational'. This is not a detail which should be left to the discussion section but should be stated upfront. You also mention who will collect the data, a detail which should appear earlier under data collection.

We have now included this in the methods section.

Line 44 – the impact of cataract is also to be examined. Again, this is new information and should not be presented for the first here.

This is added as an outcome measure in the effectiveness section

Line 49 – triangulation of different data sources mentioned but under data analysis there is no mention of how the different data will be integrated. This goes back to the suggestion to have a table clearly showing the different outcomes, the data used to assess them, and the methods used to collect these data. This would allow the reader to see much more clearly how the different data sources feed into one another to understand how the programme is working.

We have now made the data collection and data sources clearer in the table.

Line 3 (page 14) – logic models and use of theory are mentioned as strengths but again, the first time this has been mentioned. It was not made explicit that one aim is to elicit an understanding of how the intervention (or is it implementation strategy) works.

We have deleted these from this paragraph.

VERSION 2 – REVIEW

REVIEWER	Tyack, Zephania University of Queensland, Centre for Children's Burns and Trauma Research
REVIEW RETURNED	25-Dec-2020

GENERAL COMMENTS	Thank you for revising the manuscript in response to previous feedback. Some parts have been more clearly elucidated. However a few major concerns remain and clearer reporting is still required in parts of the manuscript. Several areas require major amendments in the methods: Page 8: Testing the implementation strategies separately from the effectiveness evaluation and process evaluation is part of the aim but there are few details regarding how this will be conducted. Page 11: Table 1: This table has made the outcomes clearer. However, a large number of outcomes have been chosen to measure implementation. This may make it difficult to conclude whether or not implementation has been successful. I suggest that the authors consider refining the implementation outcomes which appear to be a mixture of the implementation outcomes outlined by Proctor and the RE-AIM outcomes. It may be that either Proctor's implementation outcomes or the RE-AIM outcomes would be sufficient rather than using a combination of both. Some of the chosen outcomes seem to overlap. In addition, the rationale for choosing some of the outcomes is not clear. Adding a logic map would assist the reader in understanding how the outcomes have been used to measure the intervention implementation. Page 14: Can the investigators detail what data will be collected as part of the study for the economic evaluation? Will any QALY data be collected as part of the study? The authors mention that EQ-5D QALY utility data findings will be obtained from past studies. How will the data past study data be used in the analysis? Have the authors considered whether factors such as the age of patients seen at clinics, clinic location, or severity of diabetes impact on uptake of the DR care pathways and how will the authors consider these factors apart from including them in the regression analyses as covariates if associated with the outcome? Will the analysis of the quantitative effectiveness data be stratified by main factors that influence effectiveness? The sociodemographic details do not seem to be detailed although are likely to be captured under the quantitative data section on page 18. Could the authors make the sociodemographic details clear? Page 18: Qualitative data collection and analysis: Further detail is needed. Why will consent not be obtained? Could the authors please include a justification for the sample size and details and type of sampling that will be conducted to ensure the sample is representative of the target group (e.g., convenience, purposive).
---

	Why will only 1 health service administrator be interviewed - is there only 1 health service administrator available? What programs will be used to synthesize or code the data if any? Aspects requiring more minor revisions are: Abstract: the primary outcome in the effectiveness component needs to be more clearly represented and the abstract refined once further revisions are made to the methodology. Methods: Page 10-11: A validation test of the telemedicine approach is mentioned as a secondary analysis of acquired data. Yet the details are not specified as part of the analysis. Page 15: Can the authors please describe what the “proportion can be used in a p type control chart” refers to under ‘effectiveness’ under RE-AIM in table 1? Methods: Page 18: Statistical analysis: “Model discrimination and calibration will be assessed in order to improve transparency of the validity of the interpretation of results.” It is unclear what this statement is referring to? Does this pertain to the regression analysis? More explanation is required. The primary outcome is first referred to in the statistical analysis section. Could this be presented earlier in the methods section under outcomes? Further detail is required detailing how uptake of DR care pathway will be measured to uphold the rigour of the study. What steps have been taken to reduce bias in the measurement of this outcome? Discussion: Triangulation is discussed yet there are no specific details regarding how this will be conducted. Will divergent and convergent results be compared for the nurses, primary care doctors, ophthalmologists? Could details be added to the qualitative data collection and analysis.
--	---

REVIEWER	Riordan, Fiona University College Cork National University of Ireland, School of Public Health
REVIEW RETURNED	14-Dec-2020

GENERAL COMMENTS	Thank you for making these changes. The manuscript is improved but it still could be clearer in parts particularly in relation to the outcomes. I have made some suggestions below. When responding can you please state where in the manuscript you have made the changes (i.e., give the line number and page number in your response) as it was sometimes difficult to find the changes. I would also suggest removing page breaks and having continuous line numbers as this makes it much easier to locate changes. Line 27 – I know it is minor, but ‘eHealth’ has not been corrected The previous query about acuity was not addressed You say ‘Acceptability, feasibility, and adoption within the pilot project are tested by a staggered start of the screening program in the FHCs. We start the program with fewer number of FHCs and if more FHCs are added on to the programme it is considered as
--

	sign of acceptability and feasibility in the short term. In the long term, the adoption of the programme state-wide will be considered as success.' Yet, this is not apparent in table 1 – under acceptability there is no mention of adding FHCs. There are a couple of typos in the Article Summary: Line 19: it is not really correct to say 'the protocol will test'. Would be better to say 'this protocol describes the evaluation of a strategy to support the implementation of a new diabetic retinopathy pathway with a resource constrained environment' Line 22: Similarly, it is not a strength of the protocol but a strength of the study that x will be tested using x approach. Line 23: Would not say the 'protocol is designed to do a process evaluation' but rather a process evaluation will be conducted guided by or in line with RE-AIM framework. Line 23: should be 'integrating a diabetic retinopathy care pathway' Line 26: seems to be 'absence' twice or words missing Introduction Page 6, line 12 - Thank you for these changes. I would still suggest a citation. Page 9, line 14 - would reword as 'complex interventions of patients recruited' does not make sense. Would it be 'interventions delivered to'? I appreciate the addition of Table 1 summarising the study outcomes. This table could be organised much more clearly and consistently, particularly as it captures a large amount of information now removed from the main text. The far left column needs a label. On one on hand, it seems to organise the table content by outcome i.e., effectiveness, and later the implementation outcomes (acceptability, appropriateness, fidelity etc.) but on the other hand sometimes the content appears to be organised by strategy component (e.g. 'training of ASHAs') or by target or goal (e.g. 'assess readiness of staff'). If training is part of the implementation strategy, then by assessing the delivery of this component would that not be assessing fidelity? Some items in the 'Outcomes' column appear to be a blend of targets and methods e.g. 'increase in NCD registration' appears next to 'key informant interviews'. Is it that in some parts of the table the content of the columns changes? This make things confusing. I suggest the table be organised by outcomes and try to align with the implementation and effectiveness outcomes where you can, and given this is a type II hybrid, distinguish between outcomes relating to the evaluation of the implementation strategy and outcomes relating to the testing of the intervention, perhaps with subheadings to capture more detail. I would replace 'test' with 'method' if the intention of this column is to explain how the outcome will be assessed. In some cases the content of this column seems to be indicators so maybe that is a more fitting description. I am also confused, as, towards the end of the table, 'Process evaluation' is introduced. Would the 'implementation' component
--	---

	of RE-AIM not be informed by the other implementation outcomes mentioned earlier in the table? Also, there are two sections in the table labelled 'Effectiveness' - at start and end (under RE-AIM). Should the first instance not be 'Adoption' since this is what it appears to relate to? As this is an implementation outcome should it not be organised with the rest of those outcomes i.e., acceptability etc.? I would recommend consulting the following two articles on implementation outcomes as the definitions might help with organising the table:  • Lewis, C.C., Fischer, S., Weiner, B.J. et al. Outcomes for implementation science: an enhanced systematic review of instruments using evidence-based rating criteria. Implementation Sci • Proctor E, Silmere H, Raghavan R, Hovmand P, Aarons G, Bunger A, Griffey R, Hensley M. Outcomes for implementation research: conceptual distinctions, measurement challenges, and research agenda. Adm Policy Ment Health. The title of the table could be more informative – it details the outcomes, and how they will be assessed. I see the ERIC taxonomy is referenced in line 48 (page 7) though in relation to the evaluation approach rather than the implementation strategies. I appreciate that you have now made the distinction between the intervention and implementation strategies; however, I think it would still be helpful to more explicitly link these to the taxonomy as it might make the discrete strategies clearer. For example, is the first strategy the public awareness campaign or the improvement of the registers? If the latter, it seems this could be 'Develop and organise quality monitoring systems' according to the taxonomy – similarly point 6. My point is that the strategies could be categorised more clearly. If you have consulted the taxonomy and feel your strategies do not align with any of the 73 discrete strategies then I think that is worth stating - Powell et al acknowledge their compilation will need to be refined and possibly extended but any gaps or issues will only become evident if other researchers try to apply it. Thank you for clarifying the stakeholders in the Design section. However, you have stated 'The study protocol was developed by...'. Should it not be the DR pathway rather than the protocol? Or is it the case that these stakeholders were also involved in designing the study? If much of the detail on the process evaluation and economic evaluation has been moved now to Table 1 then you need to reference this table when first mentioning these aspects. I appreciate you have added the exact study dates to the design section. However, you have not addressed my question regarding when the proportion of patients on the register who have been screened will be assessed. Also, this is not described in Table 1 – in the first row you mention 'numbers screened' rather than proportion. You need to be consistent. Table 1. under 'Reach' in the 'test' column you mention the 'proportion offered screening' whereas in the 'outcome' column you state 'rate of participation'. Offered and participated are different so this needs to be clarified.
--	--

	I am still unsure why you say there is a 'paucity of information about NCD register size' (page 17, line 16) yet provide the size earlier (51,000) – page 10, line 31, which seems contradictory. You did not address this comment. Do you know the number of the register or is there some uncertainty here? If so, then be clear about this. Under 'Data collection' (page 18) if the relevant detail on the data is contained in Table 1 (as it appears to be) then you need to reference that table here. You have added more details of the data to be collected, however, I would still clarify which data are extracted from the register? And, also by whom or how? These details are also not in Table 1. I would still recommend reporting analyses after data collection. Is 'reading centre' (line 29, page 18) the screening centre? This is first time I think reading centre mentioned so maybe use the one term consistently. Typo in line 43 (age 18) – should this be 'premises'? Line 39 (page 18) – why will consent to participate in interview not be sought from health care professionals? I suggest providing some justification for the numbers of each professional group to be interviewed. Regarding integration, I appreciate the greater detail in table 1 but would still advise you to briefly describe your approach to integration as part of the methods section.
--	--

VERSION 2 – AUTHOR RESPONSE

Reviewer: 2

Ms. Fiona Riordan, University College Cork National University of Ireland

Competing interests of Reviewer: None declared

Comments to the Author:

Thank you for making these changes. The manuscript is improved but it still could be clearer in parts particularly in relation to the outcomes. I have made some suggestions below. When responding can you please state where in the manuscript you have made the changes (i.e., give the line number and page number in your response) as it was sometimes difficult to find the changes. I would also suggest removing page breaks and having continuous line numbers as this makes it much easier to locate changes.

Thanks and done

Line 27 – I know it is minor, but 'eHealth' has not been corrected Corrected

The previous query about acuity was not addressed-

the second effectiveness outcome is unclear; will you look at visual acuity before treatment but not after? If the intervention is effective do you expect people will present with better acuity?

We will compare presenting visual acuity of people referred to secondary care versus those obtained in the situational analysis snapshot of patients with DR self-presenting to tertiary care. We expect the visual acuity of patients from screening to have better visual acuity than self-presenting patients.

(I have highlighted this as a comment to show where it was added in the last revision in the table under effectiveness and now in Table 2 Page 13

You say 'Acceptability, feasibility, and adoption within the pilot project are tested by a staggered start of the screening program in the FHCs. We start the program with fewer number of FHCs and if more FHCs are added on to the programme it is considered as sign of acceptability and feasibility in the short term. In the long term, the adoption of the programme state-wide will be considered as success.' Yet, this is not apparent in table 1 – under acceptability there is no mention of adding FHCs.

This has been added in the row on acceptability and in the design paragraph in methods.

Line 64 abstract page 2; Table 1 in Implementation Strategy page 11 and Table 3 in Implementation Evaluation outcome page 14

There are a couple of typos in the Article Summary:

Line 19: it is not really correct to say 'the protocol will test'. Would be better to say 'this protocol describes the evaluation of a strategy to support the implementation of a new diabetic retinopathy pathway with a resource constrained environment'

Line 22: Similarly, it is not a strength of the protocol but a strength of the study that x will be tested using x approach.

Line 23: Would not say the 'protocol is designed to do a process evaluation' but rather a process evaluation will be conducted guided by or in line with RE-AIM framework.

Line 23: should be 'integrating a diabetic retinopathy care pathway'

Line 26: seems to be 'absence' twice or words missing

These are now corrected- line 76-85 page 3

Introduction

Page 6, line 12 - Thank you for these changes. I would still suggest a citation reference Done Reference 20-22

Page 9, line 14 - would reword as 'complex interventions of patients recruited' does not make sense. Would it be 'interventions delivered to'

I appreciate the addition of Table 1 summarising the study outcomes. This table could be organised much more clearly and consistently, particularly as it captures a large amount of information now removed from the main text. The far left column needs a label. On one on hand, it seems to organise the table content by outcome i.e., effectiveness, and later the implementation outcomes (acceptability, appropriateness, fidelity etc.) but on the other hand sometimes the content appears to be organised by strategy component (e.g. 'training of ASHAs') or by target or goal (e.g. 'assess readiness of staff'). If training is part of the implementation strategy, then by assessing the delivery of this component would that not be assessing fidelity?

We have now divided these into three tables – Table 1 (page 8-12) showing the implementation strategy adapted from the ERIC taxonomy as suggested. Table 2 (page 13-14) is on the effectiveness of the DR programme and Table 3(page 14-15) is the outcomes of evaluation of the implementation strategy

Some items in the 'Outcomes' column appear to be a blend of targets and methods e.g. 'increase in NCD registration' appears next to 'key informant interviews'. Is it that in some parts of the table the content of the columns changes? This make things confusing. We have now separated the outcomes on the effectiveness of the DR programme (table 2 page 13-14) and evaluation of the implementation strategy (table 3 page 14-15).

I suggest the table be organised by outcomes and try to align with the implementation and effectiveness outcomes where you can, and given this is a type II hybrid, distinguish between outcomes relating to the evaluation of the implementation strategy and outcomes relating to the testing of the intervention, perhaps with subheadings to capture more detail. Thanks and have now done this by adding Table 2 in page 13-14 and Table 3 page 14-15.

I would replace 'test' with 'method' if the intention of this column is to explain how the outcome will be assessed. It some cases the content of this column seems to be indicators so maybe that is a more fitting description. Done as Indicators and source of data Table 2 in page 13-14 and Table 3 page 14-15

I am also confused, as, towards the end of the table, 'Process evaluation' is introduced. Would the 'implementation' component of RE-AIM not be informed by the other implementation outcomes mentioned earlier in the table? We have removed RE-AIM throughout the text and the reference.

Also, there are two sections in the table labelled 'Effectiveness' - at start and end (under RE-AIM). Should the first instance not be 'Adoption' since this is what it appears to relate to? As this is an implementation outcome should it not be organised with the rest of those outcomes i.e., acceptability etc.? I would recommend consulting the following two articles on implementation outcomes as the definitions might help with organising the table:

• Lewis, C.C., Fischer, S., Weiner, B.J. et al. Outcomes for implementation science: an enhanced systematic review of instruments using evidence-based rating criteria. *Implementation Sci*

• Proctor E, Silmere H, Raghavan R, Hovmand P, Aarons G, Bunger A, Griffey R, Hensley M. Outcomes for implementation research: conceptual distinctions, measurement challenges, and research agenda. *Adm Policy Ment Health*.

We have now separated the outcomes on the effectiveness of the DR programme (Table 2 page 13-14) and evaluation of the implementation strategy (Table 3 page 14-15).

The title of the table could be more informative – it details the outcomes, and how they will be assessed.

I see the ERIC taxonomy is referenced in line 48 (page 7) though in relation to the evaluation approach rather than the implementation strategies. I appreciate that you have now made the distinction between the intervention and implementation strategies; however, I think it would still be helpful to more explicitly link these to the taxonomy as it might make the discrete strategies clearer. For example, is the first strategy the public awareness campaign or the improvement of the registers? If the latter, it seems this could be 'Develop and organise quality monitoring systems' according to the taxonomy – similarly point 6. My point is that the strategies could be categorised more clearly. If you have consulted the taxonomy and feel your strategies do not align with any of the 73 discrete strategies then I think that is worth stating - Powell et al acknowledge their compilation will need to be refined and possibly extended but any gaps or issues will only become evident if other researchers try to apply it. Thank you. We have now detailed the Implementation strategy using the ERIC taxonomy in Table 1, page 8-12.

Thank you for clarifying the stakeholders in the Design section. However, you have stated 'The study protocol was developed by...'. Should it not be the DR pathway rather than the protocol? Or is it the case that these stakeholders were also involved in designing the study? They are involved in both and this is clarified. Line 189-190 Page 6

If much of the detail on the process evaluation and economic evaluation has been moved now to Table 1 then you need to reference this table when first mentioning these aspects. We have removed process evaluation. The economic evaluation -cost-effectiveness is now in Table 2 (page 13) with reference Rachapelle et al 2013

I appreciate you have added the exact study dates to the design section. However, you have not addressed my question regarding when the proportion of patients on the register who have been screened will be assessed. The cross-sectional study will run for 12 months and then the proportions of patients screened will be assessed with denominator being numbers in the register at each FHC at start of study as we anticipate the NCD registers to increase in number of registrants as a result of training provided to ASHAs to increase public awareness. Page 18 line 386-389

Also, this is not described in Table 1 – in the first row you mention ‘numbers screened’ rather than proportion. You need to be consistent. Now changed to proportion. –see Table 2, Clinical effectiveness outcomes Page 13

Table 1. under ‘Reach’ in the ‘test’ column you mention the ‘proportion offered screening’ whereas in the ‘outcome’ column you state ‘rate of participation’. Offered and participated are different so this needs to be clarified. This is now removed

I am still unsure why you say there is a ‘paucity of information about NCD register size’ (page 17, line 16) yet provide the size earlier (51,000) – page 10, line 31, which seems contradictory. You did not address this comment. Do you know the number of the register or is there some uncertainty here? If so, then be clear about this. We know that the NCD registers have a total of 51,000 patients as of 15-03-2018. The numbers of people with diabetes in the NCD register is predicted to increase with increase awareness of this programme. We have removed this sentence and defined the denominator as the numbers in the NCD registers at each FHC at the start of the cross-sectional study.

Under ‘Data collection’ (page 18) if the relevant detail on the data is contained in Table 1 (as it appears to be) then you need to reference that table here. Added line Table 2 in line 334 and Table 3 in line 347

You have added more details of the data to be collected, however, I would still clarify which data are extracted from the register? Line 334 in page 17 from eHealth by the nurses at the FHC And, also by whom or how? Anonymised data as serial number onto study database These details are also not in Table 1. Data sources are shown to Table 2 and 3.

I would still recommend reporting analyses after data collection. Done

Is ‘reading centre’ (line 29, page 18) the screening centre? This is first time I think reading centre mentioned so maybe use the one term consistently. It is now mentioned in the section on Evidence-based clinical intervention Line 217 Page 7

Typo in line 43 (age 18) – should this be ‘premises’? corrected

Line 39 (page 18) – why will consent to participate in interview not be sought from health care professionals? Sorry I mean no written consent. This is now clarified as Verbal consent Line 353 page 17

I suggest providing some justification for the numbers of each professional group to be interviewed. Done -line 350-353 Page 17.

Regarding integration, I appreciate the greater detail in table 1 but would still advise you to briefly describe your approach to integration as part of the methods section. This is now included in Table 1 page 8 and the outcome in Table 3 page 16.

Reviewer: 1

Dr. Zephania Tyack, University of Queensland

Competing interests of Reviewer: None declared

Comments to the Author:

Thank you for revising the manuscript in response to previous feedback. Some parts have been more clearly elucidated. However a few major concerns remain and clearer reporting is still required in parts of the manuscript.

Several areas require major amendments in the methods:

Page 8: Testing the implementation strategies separately from the effectiveness evaluation and process evaluation is part of the aim but there are few details regarding how this will be conducted.

We have now detailed the Implementation strategy in table 1 and we have separated the effectiveness outcome and the implementation evaluation outcome in table 2 and 3, respectively.

Page 11: Table 1: This table has made the outcomes clearer. However, a large number of outcomes have been chosen to measure implementation. This may make it difficult to conclude whether or not implementation has been successful. I suggest that the authors consider refining the implementation outcomes which appear to be a mixture of the implementation outcomes outlined by Proctor and the RE-AIM outcomes. It may be that either Proctor's implementation outcomes or the RE-AIM outcomes would be sufficient rather than using a combination of both. Some of the chosen outcomes seem to overlap. We have now removed the RE-AIM outcomes and used the Proctor's implementation outcomes.

In addition, the rationale for choosing some of the outcomes is not clear. Adding a logic map would assist the reader in understanding how the outcomes have been used to measure the intervention implementation. A logic model is now added as Figure 2.

Page 14: Can the investigators detail what data will be collected as part of the study for the economic evaluation? Will any QALY data be collected as part of the study? The authors mention that EQ-5D QALY utility data findings will be obtained from past studies. How will the data past study data be used in the analysis?

The economic analysis uses data that will be collected as part of the study in Kerala on:

- number of people screened and number of screenings,
- number of people referred and number of ungradable images,
- number of people diagnosed with STDR and number diagnosed with cataract,
- average age of those treated for cataract and of those treated for STDR.

The data collection from patients screened in the FHCs will not include quality of life data other than a question on life satisfaction. We anticipate that it will not be possible to collect large amounts of data from patients due to time restraints on FHC staff. There is a strict limit to the amount of data that could be collected at FHC screening sessions, which have to see large numbers of patients quickly.

For these reasons we use utility data from Rachapelle et al (2013,) table 2, which shows 0.87 for no DR, 0.79 for non-STDR DR, 0.70 for STDR and 0.55 for bilateral blindness. (Rachapelle, S., Legood, R., Alavi, Y., Lindfield, R., Sharma, T., Kuper, H., & Polack, S. (2013). The cost–utility of telemedicine to screen for diabetic retinopathy in India. *Ophthalmology*, 120(3), 566-573.)

We use the EQ-5D utility data findings from Rachapelle et al to ‘weight’ the remaining life years of those treated for STDR and for people not treated for it. More specifically –

- We are developing a Markov model using data derived from the literature on transition rates between STDR, blindness and death.
- We are using this model to estimate total and quality-adjusted life expectancy (QALE), at the average age of those treated, both if STDR is untreated and if it is treated: we assume that treatment stops the process of decline to blindness and that, in the context of India, no screening programme generally means no treatment and ultimately loss of sight.
- In estimating quality-adjusted life expectancy, we are applying the utility findings from Rachapelle et al to the expected numbers of years in each state, e.g. years with STDR are weighted by 0.70 and years without STDR (after treatment) by 0.87.
- We are discounting future years using a discount rate of 3% and take the difference between QALE with treatment and QALE without treatment, which provides our estimate of (discounted) lifetime QALY gain from treatment.

We are using a broadly similar approach for cataract but drawing on further material from past studies. We consider that we should include cataract surgery in our analysis, although the screening programme relates to DR, since for the patients concerned it is the screening programme which leads to their referral, diagnosis of cataract and treatment for cataract.

We are estimating the cost to the Government of Kerala and the societal cost of the screening programme, using a range of data collected in Kerala, data provided by the wider research team from their hospitals and expert opinion of clinical members of the research team. We are dividing the total cost of the programme by the number of patients treated for STDR or cataract to estimate the average cost per person treated of the screening programme. To this we add the cost of treatment.

We are estimating the incremental cost-effectiveness ratio (ICER) in the usual way and conducting sensitivity analyses.

Have the authors considered whether factors such as the age of patients seen at clinics, clinic location, or severity of diabetes impact on uptake of the DR care pathways and how will the authors consider these factors apart from including them in the regression analyses as covariates if associated with the outcome?

Thank you for raising this point. We will use the evidence we get from studying the relationship of the factors like age, location, severity of diabetes, and others to produce a policy briefing for the Government of Kerala on strategies for sustainability and upscaling of the program.

Will the analysis of the quantitative effectiveness data be stratified by main factors that influence effectiveness? Yes

The sociodemographic details do not seem to be detailed although are likely to be captured under the quantitative data section on page 18. Could the authors make the sociodemographic details clear?
Added in line 335-page 17

Page 18: Qualitative data collection and analysis: Further detail is needed. Why will consent not be obtained? Sorry, we meant to write no written consent. This is now clarified as Verbal consent Line 353 page 17

Could the authors please include a justification for the sample size and details and type of sampling that will be conducted to ensure the sample is representative of the target group (e.g., convenience, purposive). Why will only 1 health service administrator be interviewed - is there only 1 health service administrator available? What programs will be used to synthesize or code the data if any?

A pragmatic approach is taken. We intend to request all available FHC nurses if they would be willing to participate in the qualitative study but we believe it will be a convenient sample from the target group. The administrator is the Health Secretary himself. The qualitative data will be coded using NVivo and analysed using thematic analysis following the strategy.

Aspects requiring more minor revisions are:

Abstract: the primary outcome in the effectiveness component needs to be more clearly represented and the abstract refined once further revisions are made to the methodology. Done line 57 Page 2

Methods: Page 10-11: A validation test of the telemedicine approach is mentioned as a secondary analysis of acquired data. Yet the details are not specified as part of the analysis. This is now written in the table 3 under Fidelity and in statistical analysis on page 424-426

Page 15: Can the authors please describe what the “proportion can be used in a p type control chart” refers to under ‘effectiveness’ under RE-AIM in table 1? We will then calculate (1-number of patients screened)/number of patients that can be screened). We have now added reference 28.

Methods: Page 18: Statistical analysis: “Model discrimination and calibration will be assessed in order to improve transparency of the validity of the interpretation of results.” It is unclear what this statement is referring to? Does this pertain to the regression analysis? More explanation is required.

We have edited the sentence to make it clearer. Line 423-425

The primary outcome is first referred to in the statistical analysis section. Could this be presented earlier in the methods section under outcomes? It is now in the abstract line 57 and table 2.

Further detail is required detailing how uptake of DR care pathway will be measured to uphold the rigour of the study. What steps have been taken to reduce bias in the measurement of this outcome? The uptake of DR care pathway will be measured by a telephone survey of all patients who are referred to secondary and tertiary care centre (i.e. enrolled into the DR care pathway)-see Table 3. We will try to reduce the bias in measurement by including all eligible patients. However, it is possible that we may not be able to get everyone to respond, in which case we will quantify that bias and mitigate it in the analysis phase.

Discussion: Triangulation is discussed yet there are no specific details regarding how this will be conducted. Will divergent and convergent results be compared for the nurses, primary care doctors, ophthalmologists? This was a mistake. Changed to “use” in line 570-571 page 21

Could details be added to the qualitative data collection and analysis. We have mentioned in Table 3 and in the statistical analysis strategy. Line 428-431 page 19. Data will be coded to identify themes that the researchers consider pertinent to the study outcomes. The entire dataset will be given equal attention so that full consideration for an iterative data handling process will be undertaken to develop and tabulate into set of themes.

VERSION 3 – REVIEW

REVIEWER	Tyack, Zephania University of Queensland, Centre for Children's Burns and Trauma Research
REVIEW RETURNED	13-Feb-2021

GENERAL COMMENTS	The authors have been responsive to most of my prior comments but a few major and minor revisions are still needed, as follows: Major revision Table 1 includes a large number of implementation strategies that may make it difficult to evaluate which strategies contributed most to the effectiveness and implementation. The staggered implementation with phasing in of five FHCs starting as pilots to enable small changes before gradually moving to other 11 FHCs may create evaluation complexities in that the dose of the intervention and implementation strategies may differ over time at multiple levels (patient, clinician, FHC). How will this be monitored and accounted for? I am assuming the multivariable regression models will be able to account for this variation but it would be good to make an explicit statement regarding this in the methods. The discussion indicates that results of the study will inform the adoption of this pathway in other areas in India and globally. However, the complexity and number of strategies, and lack of validated implementation outcomes may limit generalisability of the results and implementation of the pathway elsewhere and should be acknowledged as a limitation. Table 3: The acceptability outcome is not 100% clear. Can you be more explicit in explaining the data source, which presently is stated very broadly as 'clinical performance from FHCs?' It would also be good to add an explicit statement regarding statewide adoption being considered a success which could be added to this table or elsewhere in the text, as per reviewer 1 comments. It is unclear how "Data source: Policy paper to scale-up to other districts" relates to adoption. Can more detail be added? Minor revisions As per Reviewer 1's comments some statements referring to the protocol would be better to refer to the study. For example, in one of the article summary points "The protocol is limited by the absence of a comparator due to lack of previous data on the prevalence of DR in the public health system." it would be better to replace the word protocol with study. Please check this throughout the manuscript. Introduction Last sentence: "The transformation of primary with a focus on NCDs provided the backdrop to the implementation of a DR care pathway." Is the word 'care' missing after primary? Methods: Aims and objectives: To be consistent with the logic model it would be good to reword the aim slightly to include patient/ clinician and service level outcomes. For example, the proposed study will (1) examine the clinical and cost effectiveness of 185 the DR care pathway at the patient/clinician and service levels and (2) evaluate the implementation strategy of the pathway. Setting: last sentence: insert a full stop. Implementation strategies section: check punctuation as there are errors in this section. Quantitative data: further detail is needed regarding how outcomes such as vision related quality of life will be measured? Will a
---

	validated questionnaire be used? Details could be added in a supplementary file and briefly mentioned in the text. Qualitative data collection and analysis: Monitoring: The authors state the “Only anonymised extracted data from eHealth will be used for analysis.”. This statement seems out of place under a qualitative sub heading. What qualitative data will be extracted from eHealth for the qualitative data analysis? If patient qualitative data is entered into the electronic medical record this might be a barrier to issues being reported. It may be that the authors intend the monitoring heading to be a major heading encompassing qualitative and quantitative data – in this case the heading should be changed and the authors need to be clearer adding details regarding what qualitative and quantitative data monitoring entails. Could the authors also remove the abbreviation eHealth and replace with electronic medical record for better understanding throughout the manuscript? Discussion: First sentence: “At the conclusion of this study, we hope to assess the feasibility, effectiveness and implementation challenges of a complex DR care pathway integrating care at primary, secondary and tertiary care, covering a proportion of the diabetic population in Thiruvananthapuram.” Feasibility is captured under implementation thus does not need to be mentioned separately as this will confuse readers. Consistency in terminology should be checked throughout the manuscript.
--	--

REVIEWER	Riordan, Fiona University College Cork National University of Ireland, School of Public Health
REVIEW RETURNED	02-Feb-2021

GENERAL COMMENTS	Thank you for making these changes. There are still a few revisions required. ‘Acceptability, feasibility, and adoption within the pilot project are tested by a staggered start of the screening program in the FHCs. We start the program with fewer number of FHCs and if more FHCs are added on to the programme it is considered as sign of acceptability and feasibility in the short term. In the long term, the adoption of the programme state-wide will be considered as success.’ I see the changes in Table 3 (though not highlighted). I can see no highlighted changes in line 64 Abstract (apart from attention of word ‘penetration’) and Table 1 Implementation Strategy page 11 even though these were specified in the author’s response, so I am less clear what was amended here. I think the addition to Table 3 addresses the comment but it would be important to know what further changes were made. Would suggest using bold text or spacing to distinguish the data sources from the description of the outcomes within the third column of Table 3. Also keep the ordering consistent. At the moment this information is lost. There are still errors line 76-85. For example, the protocol does not describe effectiveness. This is a protocol - my understanding is that the study has not been completed yet and therefore effectiveness has not yet been tested.
--

	It does not make sense to say the 'type 2 hybrid effectiveness-implementation will be tested using...' as this is the trial design. More correct to say it will use or employ mixed methods. Line 176: would be better to say: 'using both qualitative and quantitative methods' given you have now included the word 'evaluate'. Thank you for the addition of Tables 2 and 3, I think this makes things much clearer. One suggestion would be to keep the ordering consistent; so, in Table 2 you have data source first and indicators to the far right whereas in Table 3, data source is in far right column. I feel it makes more sense to present the indicator first then the data source (as done for Table 3) and have this consistent in both tables. In Table 3 under 'Source of data and assessment' some information seems to correspond to what was classified under 'indicator' in the Table 2, for example 'Proportions consenting to be screened'. In this table it seems as though the second column instead presents subcategories or components (or dimensions?) of the implementation outcome rather than indicators. For example, under Adoption, it reports uptake by staff, patients, and in settings FHC secondary and tertiary. Under Appropriateness, it reports different aspects of appropriateness, be it referrals to secondary care or the primary-secondary care pathway. I appreciate it is difficult to organise given the large amount of data being collect and the number of outcomes, but I think this table could benefit from a little more formatting (see comment above about bolding/spacing) and considering the column labels. I cannot see the piece about the cross-sectional study running for 12 months on page 18 line 386-389 as stated. There is no line 386 page 18? It may be stated elsewhere but I could not see it. Line 334, page 17 I see no reference to extraction from eHealth by nurses. I think the line and page numbering may be wrong, but I could not find this reference in the rest of document. I cannot see any edits in Table 1 or Table 3 relating to integration. Could you please clarify what text has been changed? It seems more appropriate to describe your approach in the Methods rather than the outcome tables.
--	---

VERSION 3 – AUTHOR RESPONSE

Reviewer: 2

Ms. Fiona Riordan, University College Cork National University of Ireland

Comments to the Author:

Thank you for making these changes. There are still a few revisions required.

'Acceptability, feasibility, and adoption within the pilot project are tested by a staggered start of the screening program in the FHCs. We start the program with fewer number of FHCs and if more FHCs are added on to the programme it is considered as sign of acceptability and feasibility in the short term. In the long term, the adoption of the programme state-wide will be considered as success.'

I see the changes in Table 3 (though not highlighted). I can see no highlighted changes in line 64 Abstract (apart from attention of word 'penetration') and Table 1 Implementation Strategy page 11 even though these were specified in the author's response, so I am less clear what was amended here. I think the addition to Table 3 addresses the comment but it would be important to know what further changes were made.

The abstract only allows 300 words. We have now removed the sentences on the comparator (63-65) to add these sentences. Lines 67-69

We have added all the measures for evaluating implementation strategy in Table 1 highlighted and re-arranged table 3 as per another comment below.

Would suggest using bold text or spacing to distinguish the data sources from the description of the outcomes within the third column of Table 3. Also keep the ordering consistent. At the moment this information is lost.

Ordering and bold texting done as advised. Please see tracked changes in Table 3, page 16 onwards.

There are still errors line 76-85.

For example, the protocol does not describe effectiveness. This is a protocol - my understanding is that the study has not been completed yet and therefore effectiveness has not yet been tested.

We have edited these sentences to reflect these suggestions Lines 86-90.

It does not make sense to say the 'type 2 hybrid effectiveness-implementation will be tested using...' as this is the trial design. More correct to say it will use or employ mixed methods.

We have now changed the title of the manuscript and line 46 in the Abstract and line 198 to reflect this.

Line 176: would be better to say: 'using both qualitative and quantitative methods' given you have now included the word 'evaluate'.

Done Line 186

Thank you for the addition of Tables 2 and 3, I think this makes things much clearer. One suggestion would be to keep the ordering consistent; so, in Table 2 you have data source first and indicators to the far right whereas in Table 3, data source is in far right column. I feel it makes more sense to present the indicator first then the data source (as done for Table 3) and have this consistent in both tables.

We have now moved the columns as suggested and please see track changes in Table 2.

In Table 3 under 'Source of data and assessment' some information seems to correspond to what was classified under 'indicator' in the Table 2, for example 'Proportions consenting to be screened'. In this table it seems as though the second column instead presents subcategories or components (or dimensions?) of the implementation outcome rather than indicators. For example, under Adoption, it reports uptake by staff, patients, and in settings FHC secondary and tertiary. Under Appropriateness, it reports different aspects of appropriateness, be it referrals to secondary care or the primary-secondary care pathway. I appreciate it is difficult to organise given the large amount of data being collect and the number of outcomes, but I think this table could benefit from a little more formatting (see comment above about bolding/spacing) and considering the column labels.

We have tried to edit Table 3 as suggested- please see track changes

I cannot see the piece about the cross-sectional study running for 12 months on page 18 line 386-389 as stated. There is no line 386 page 18? It may be stated elsewhere but I could not see it.

I am sorry about this. I worked on a Mac and after receiving your comments, I realised that the numbering on tracked version jumped by a few lines at each page-break in my Mac and so exceeded

the numbers others would see on the same document. I am unable to correct this so another co-author has done the line number on her computer this time.
We have written it in abstract line 65, main manuscript line 206-207.

Line 334, page 17 I see no reference to extraction from eHealth by nurses. I think the line and page numbering may be wrong, but I could not find this reference in the rest of document.
We have added this in line 305.

I cannot see any edits in Table 1 or Table 3 relating to integration. Could you please clarify what text has been changed? It seems more appropriate to describe your approach in the Methods rather than the outcome tables.

We have highlighted in yellow all areas relating to Integration.

Reviewer: 1

Dr. Zephania Tyack, University of Queensland

Comments to the Author:

The authors have been responsive to most of my prior comments but a few major and minor revisions are still needed, as follows:

Major revision

Table 1 includes a large number of implementation strategies that may make it difficult to evaluate which strategies contributed most to the effectiveness and implementation. The staggered implementation with phasing in of five FHCs starting as pilots to enable small changes before gradually moving to other 11 FHCs may create evaluation complexities in that the dose of the intervention and implementation strategies may differ over time at multiple levels (patient, clinician, FHC). How will this be monitored and accounted for? I am assuming the multivariable regression models will be able to account for this variation but it would be good to make an explicit statement regarding this in the methods.

Thank you for raising this point. We are aware of the complexities a staggered approach will add to the study and we hoped to take care of it in the regression models. A non-significant coefficient for this variable will suggest the staggered approach had no effect and a significant coefficient will allow the effect to be quantified for each FHC. We have added a sentence to indicate this at line 378-383.

The discussion indicates that results of the study will inform the adoption of this pathway in other areas in India and globally. However, the complexity and number of strategies, and lack of validated implementation outcomes may limit generalisability of the results and implementation of the pathway elsewhere and should be acknowledged as a limitation.

We have added this as a limitation in lines 431-433

Table 3: The acceptability outcome is not 100% clear. Can you be more explicit in explaining the data source, which presently is stated very broadly as 'clinical performance from FHCs?'

We have added more details on data source in this row in Table 3.

It would also be good to add an explicit statement regarding statewide adoption being considered a success which could be added to this table or elsewhere in the text, as per reviewer 1 comments.

This is added in abstract line 69-70 and Table 3 under Adoption

It is unclear how "Data source: Policy paper to scale-up to other districts" relates to adoption. Can more detail be added?

This is added in the row on Adoption in Table 3 page 17/18.

Minor revisions

As per Reviewer 1's comments some statements referring to the protocol would be better to refer to the study. For example, in one of the article summary points "The protocol is limited by the absence of a comparator due to lack of previous data on the prevalence of DR in the public health system." it would be better to replace the word protocol with study. Please check this throughout the manuscript. 'Protocol' replaced with 'study' throughout manuscript except the title of paper as previously advised by the Editorial Board

Introduction

Last sentence: "The transformation of primary with a focus on NCDs provided the backdrop to the implementation of a DR care pathway." Is the word 'care' missing after primary?

Added "care" line 158

Methods:

Aims and objectives: To be consistent with the logic model it would be good to reword the aim slightly to include patient/ clinician and service level outcomes. For example, the proposed study will (1) examine the clinical and cost effectiveness of 185 the DR care pathway at the patient/clinician and service levels and (2) evaluate the implementation strategy of the pathway.

Added line 194

Setting: last sentence: insert full stop.

Full stop added line 230

Implementation strategies section: check punctuation as there are errors in this section.

The table is edited for punctuations.

Quantitative data: further detail is needed regarding how outcomes such as vision related quality of life will be measured? Will a validated questionnaire be used? Details could be added in a supplementary file and briefly mentioned in the text.

We have added the reference for EQ-5D Bolt-on (line 296-297 and reference 29) and added it as a supplementary file 1.

Qualitative data collection and analysis: Monitoring: The authors state the "Only anonymised extracted data from eHealth will be used for analysis.". This statement seems out of place under a qualitative sub heading. What qualitative data will be extracted from eHealth for the qualitative data analysis? If patient qualitative data is entered into the electronic medical record this might be a barrier to issues being reported. It may be that the authors intend the monitoring heading to be a major heading encompassing qualitative and quantitative data – in this case the heading should be changed and the authors need to be clearer adding details regarding what qualitative and quantitative data monitoring entails.

Moved the data monitoring section to sub section under Quantitative data lines 303 onwards

Could the authors also remove the abbreviation eHealth and replace with electronic medical record for better understanding throughout the manuscript?

Changed to electronic health records (EHR) throughout manuscript

Discussion: First sentence: "At the conclusion of this study, we hope to assess the feasibility, effectiveness and implementation challenges of a complex DR care pathway integrating care at primary, secondary and tertiary care, covering a proportion of the diabetic population in Thiruvananthapuram." Feasibility is captured under implementation thus does not need to be mentioned separately as this will confuse readers.

'Feasibility' removed from line 404

Consistency in terminology should be checked throughout the manuscript.

Checked throughout manuscript.

VERSION 4 – REVIEW

REVIEWER	Tyack, Zephania University of Queensland, Centre for Children's Burns and Trauma Research
REVIEW RETURNED	02-Apr-2021

GENERAL COMMENTS	The authors have been responsive to most of the feedback previously provided. Some minor amendments are still needed. General comments There are too many abbreviations some of which are not commonly known (e., SDGs, NCDs, FHCs, RIO). Please reduce the abbreviations and ensure any remaining abbreviations are commonly known and spelt out in full the first time used in the text. Abstract A sentence on the study analysis should be added. Table 1 implementation strategies Specific actions column: (1) “This will serve as the comparator for the effectiveness outcome.” It is not clear what this sentence refers to – could some further detail be added? (2) “In addition to integration, it will contribute to both implementation outcomes.” Please state the implementation outcomes being referred to here. Qualitative data collection and analysis The reference to thematic analysis is not needed as the authors have mentioned using a descriptive phenomenological approach which should be briefly described. Outcomes Line 296: The authors mention the EQ-5D-5L-Vision Bolt-on for the first time. It is unclear how this measure is to be used in the evaluation. Is this the measure that will be used to measure QALYs, if not please also indicate which measure is used to measure QALYs? Please also provide some brief details regarding the validation of this measure (e.g., content validation, population used for validation, structural validity). Please ensure that you have permission to reproduce the detail on the EQ-5D-5L-Vision Bolt-on in Supplementary file 1 and include this in the supplementary file. Usually such a measure is protected by copyright and thus cannot be reproduced without the permission of the publishers/ authors. However a brief description of the items and scales can be provided in the text. The reference 29 which has been applied to the measure doesn't appear to be the original reference for the measure. Could the original reference please be added? Statistical analyses Line 386: “The validation of the telemedicine will be reported as the agreement (kappa statistics) between screen positive patients graded by the graders at the reading centre versus the DR grade as recorded by the ophthalmologists in secondary care centres.” What kappa coefficient is pre-specified to indicate validity? Data monitoring section
--

	This section doesn't fit well after the quantitative data heading as it applies to both the qualitative and quantitative data. I think it may be best after both the qualitative and quantitative headings as it refers to coding which is clearly qualitative data. Discussion The length of the discussion could be reduced as there is some repetition. Some statements need to be made less definitive as the findings are not yet known. For example, local contextual factors mean the findings may not generalise to other settings so the statement about generalisability needs to be tempered. Line 405: "At the conclusion of this study, we hope to assess the effectiveness and implementation challenges...". I suggest replacing the word 'challenges' with 'outcomes' as you will assess the barriers as well as enablers. Other statements to be written less definitively are: Line 412: "Early identification of STDR and other complications due to this holistic approach and timely treatment, will have positive impact on rates of blindness, chronic kidney disease, cardiovascular complications and thereby improve health, reduce multi-morbidity and mortality." : replace 'will' with 'are expected to'. Line 415: " A DR pathway that straddles primary, secondary and tertiary levels of care, leveraging technology will have advantages of cost effectiveness and ease of implication in LMICs as against the current practice of detection and management of self-reported cases in tertiary centres. ": replace 'will' with 'may be'; and reword 'ease of implication'. Line 431: "The results of this study will inform the adoption of this pathway in other areas in India and globally." should be revised to: "The results of this study may inform the adoption of this pathway in other areas in India and globally." Line 432: "However, the complexity and number of implementation strategies and lack of validated implementation outcomes may limit generalisability of the results and implementation of this pathway elsewhere.": contextual factors should also be referred to in this sentence. Line 452: "This study has many strengths": I suggest removing the word 'many'. Supplementary file 2: consent form: punctuation needs to be corrected.
--	--

REVIEWER	Riordan, Fiona University College Cork National University of Ireland, School of Public Health
REVIEW RETURNED	15-Mar-2021

GENERAL COMMENTS	Thank you for these changes. Table 1 describes the implementation strategies – I do not think it is necessary or makes sense to specify which implementation outcome will be assessed under each element of the strategies; i.e., the last sentences in each section starting with 'This will contribute to...'. I would remove these additions and keep this table just as a description of the strategies. Apologies for any confusion. My comment on the last version was simply a query to clarify what had been changed 'Table 1 page 11' as this was
---

	mentioned in the previous response, but no tracked changes were visible. If no changes were made and this was cited in error that is fine. My suggestion regarding the reference to type 2 hybrid effectiveness-implementation design, related to the phrasing in line 91 rather than the use of this term. So, I would suggest just rephrasing rather than omitting this description entirely, unless this is no longer the design being used. Therefore, in the Abstract I think you should still mention that you are using a type 2 hybrid effectiveness-implementation design. So, for example, in the first line you could say ‘Using a type 2 hybrid effectiveness-implementation design, we aim to pilot a diabetic retinopathy (DR) care pathway in the public health system in the Thiruvananthapuram district in Kerala to understand how it can be scaled up to and sustained in the whole state.’ In line 91 could rephrase to ‘The study will use a mixed methods type 2- hybrid effectiveness-implementation design.’ In line 198 could rephrase to ‘We chose a mixed methods type 2- hybrid effectiveness-implementation design to evaluate the effectiveness of the clinical interventions and the implementation strategies.’ Line 59 – I think this should this read ‘over’ rather than ‘by’? So, ‘The primary effectiveness outcome is the proportion of patients in the NCD register with 60 diabetes screened for DR over 12 months.’
--	---

VERSION 4 – AUTHOR RESPONSE

Reviewer 2: Ms. Fiona Riordan, University College Cork National University of Ireland

Table 1 describes the implementation strategies – I do not think it is necessary or makes sense to specify which implementation outcome will be assessed under each element of the strategies; i.e., the last sentences in each section starting with ‘This will contribute to...’. I would remove these additions and keep this table just as a description of the strategies. Apologies for any confusion. My comment on the last version was simply a query to clarify what had been changed ‘Table 1 page 11’ as this was mentioned in the previous response, but no tracked changes were visible. If no changes were made and this was cited in error that is fine.

We have now removed all the implementation outcomes from Table 1.

My suggestion regarding the reference to type 2 hybrid effectiveness-implementation design, related to the phrasing in line 91 rather than the use of this term. So, I would suggest just rephrasing rather than omitting this description entirely, unless this is no longer the design being used.

Therefore, in the Abstract I think you should still mention that you are using a type 2 hybrid effectiveness-implementation design.

So, for example, in the first line you could say ‘Using a type 2 hybrid effectiveness-implementation design, we aim to pilot a diabetic retinopathy (DR) care pathway in the public health system in the Thiruvananthapuram district in Kerala to understand how it can be scaled up to and sustained in the whole state.’

Added line 44

In line 91 could rephrase to 'The study will use a mixed methods type 2- hybrid effectiveness-implementation design.'

Added line 84-85

In line 198 could rephrase to 'We chose a mixed methods type 2- hybrid effectiveness-implementation design to evaluate the effectiveness of the clinical interventions and the implementation strategies.'

Added line 193

Line 59 – I think this should read 'over' rather than 'by'? So, 'The primary effectiveness outcome is the proportion of patients in the NCD register with 60 diabetes screened for DR over 12 months.'

Corrected line 61

Reviewer: 1

Dr. Zephania Tyack, University of Queensland

General comments

There are too many abbreviations some of which are not commonly known (e.g., SDGs, NCDs, FHCs, RIO). Please reduce the abbreviations and ensure any remaining abbreviations are commonly known and spelt out in full the first time used in the text.

We have now spelt out most of the abbreviations to reduce the number of abbreviations.

Abstract

A sentence on the study analysis should be added.

Added line 58-60, but 2 previous sentences removed due to word limit.

Table 1 implementation strategies

Specific actions column:

(1) "This will serve as the comparator for the effectiveness outcome." It is not clear what this sentence refers to – could some further detail be added?

(2) "In addition to integration, it will contribute to both implementation outcomes." Please state the implementation outcomes being referred to here.

Reviewer 2 has asked that these be removed from Table 1 which we have now done.

Qualitative data collection and analysis

The reference to thematic analysis is not needed as the authors have mentioned using a descriptive phenomenological approach which should be briefly described.

Added line 324-327

Outcomes

Line 296: The authors mention the EQ-5D-5L-Vision Bolt-on for the first time. It is unclear how this measure is to be used in the evaluation. Is this the measure that will be used to measure QALYs, if not please also indicate which measure is used to measure QALYs? Please also provide some brief details regarding the validation of this measure (e.g., content validation, population used for validation, structural validity).

We have added this is for QALY.

Please ensure that you have permission to reproduce the detail on the EQ-5D-5L-Vision Bolt-on in Supplementary file 1 and include this in the supplementary file. Usually such a measure is protected by copyright and thus cannot be reproduced without the permission of the publishers/ authors.

I am one of the authors of the clinical trial and developed the EQ-5D-5L Vision Bolt worksheets and the clinical trial group is aware and approved its use for other studies. Euroqol does not require permission for non-commercial research line286-293.

However a brief description of the items and scales can be provided in the text. The reference 29 which has been applied to the measure doesn't appear to be the original reference for the measure. Could the original reference please be added?
These have been added -references 30 and 31.

Statistical analyses

Line 386: "The validation of the telemedicine will be reported as the agreement (kappa statistics) between screen positive patients graded by the graders at the reading centre versus the DR grade as recorded by the ophthalmologists in secondary care centres." What kappa coefficient is pre-specified to indicate validity?

Now added line 390-391

Data monitoring section

This section doesn't fit well after the quantitative data heading as it applies to both the qualitative and quantitative data. I think it may be best after both the qualitative and quantitative headings as it refers to coding which is clearly qualitative data.

Moved as suggested

.

Discussion

The length of the discussion could be reduced as there is some repetition.

Some statements need to be made less definitive as the findings are not yet known. For example, local contextual factors mean the findings may not generalise to other settings so the statement about generalisability needs to be tempered.

Deleted some sentences from discussion

Line 405: "At the conclusion of this study, we hope to assess the effectiveness and implementation challenges...". I suggest replacing the word 'challenges' with 'outcomes' as you will assess the barriers as well as enablers.

Amended line 408

Other statements to be written less definitively are:

Line 412: "Early identification of STDR and other complications due to this holistic approach and timely treatment, will have positive impact on rates of blindness, chronic kidney disease, cardiovascular complications and thereby improve health, reduce multi-morbidity and mortality." : replace 'will' with 'are expected to'.

Amended line 415

Line 415: " A DR pathway that straddles primary, secondary and tertiary levels of care, leveraging technology will have advantages of cost effectiveness and ease of implication in LMICs as against the current practice of detection and management of self-reported cases in tertiary centres. ": replace 'will' with 'may be'; and reword 'ease of implication'.

Amended line 418-419

Line 431: "The results of this study will inform the adoption of this pathway in other areas in India and globally." should be revised to: "The results of this study may inform the adoption of this pathway in other areas in India and globally."

Amended line 434

Line 432: “However, the complexity and number of implementation strategies and lack of validated implementation outcomes may limit generalisability of the results and implementation of this pathway elsewhere.”: contextual factors should also be referred to in this sentence.
 Added line 435

Line 452: “This study has many strengths”: I suggest removing the word ‘many’.
 This sentence has been reworded-line 456

Supplementary file 2: consent form: punctuation needs to be corrected.
 Corrected

VERSION 5 – REVIEW

REVIEWER	Tyack, Zephania University of Queensland, Centre for Children's Burns and Trauma Research
REVIEW RETURNED	02-May-2021

GENERAL COMMENTS	The manuscript is much improved with some minor issues remaining. As indicated by reviewer 2, type 2 hybrid effectiveness-implementation design should be used to refer to the design and mixed methods as the method of evaluation. I think the use of mixed methods is more important to point out in the article summary than the design which is in the title. As per reviewer 2's comments, the line numbers the authors have cited to indicate changes do not seem to be correct so it is not clear where may of the changes have been made for revision 4. For example, the following statements by the authors do not seem to align with the quoted line numbers: “We have added a sentence to indicate this at line 378-383”, “We have added this as a limitation in lines 431-433”. Qualitative analysis: line 325-328: “We will transcribe all the participants’ descriptions, extract significant statements or phrases, create formulated meanings or meaning units, aggregate formulated meanings and incorporate the result into descriptions.” This sentence should be reworded slightly to refer to transcribing interview data rather than transcribing the participants descriptions. To my knowledge ‘extract’ is not a term usually applied to this analysis and ‘significant’ is a term that is best reserved for statistics in an academic paper. Is descriptive phenomenological analysis appropriate for field note analysis as in the statement: “Field notes, the interviews and focus group content will provide the basis for the data analysis, which will be based on a descriptive phenomenological approach without data or opinion interpretation and will include transcription, condensation, coding and categorisation using qualitative analysis tools.” I would argue field note data is used to inform the analysis of interview data but does not directly have a phenomenological approach applied. See the following paper on Qualitative Research: https://www.ncbi.nlm.nih.gov/pmc/articles/PMC4485510/
---

VERSION 5 – AUTHOR RESPONSE

Reviewer: 1

Dr. Zephania Tyack, University of Queensland

Comments to the Author:

The manuscript is much improved with some minor issues remaining.

As indicated by reviewer 2, type 2 hybrid effectiveness-implementation design should be used to refer to the design and mixed methods as the method of evaluation. I think the use of mixed methods is more important to point out in the article summary than the design which is in the title.

This has been amended in the article summary and the Methods section.

As per reviewer 2's comments, the line numbers the authors have cited to indicate changes do not seem to be correct so it is not clear where may of the changes have been made for revision 4. For example, the following statements by the authors do not seem to align with the quoted line numbers: "We have added a sentence to indicate this at line 378-383", "We have added this as a limitation in lines 431-433".

We have tidied up the numbering again.

Qualitative analysis: line 325-328: "We will transcribe all the participants' descriptions, extract significant statements or phrases, create formulated meanings or meaning units, aggregate formulated meanings and incorporate the result into descriptions."

This sentence should be reworded slightly to refer to transcribing interview data rather than transcribing the participants descriptions. To my knowledge 'extract' is not a term usually applied to this analysis and 'significant' is a term that is best reserved for statistics in an academic paper.

We have edited these sentences.-see lines 312-313

Is descriptive phenomenological analysis appropriate for field note analysis as in the statement:

"Field notes, the interviews and focus group content will provide the basis for the data analysis, which will be based on a descriptive phenomenological approach without data or opinion interpretation and will include transcription, condensation, coding and categorisation using qualitative analysis tools."

I would argue field note data is used to inform the analysis of interview data but does not directly have a phenomenological approach applied. See the following paper on Qualitative

Research:<https://www.ncbi.nlm.nih.gov/pmc/articles/PMC4485510/>

Thank you for pointing out our mistake in including field notes in the material used for analysis using phenomenological approach. Field notes help in understanding the phenomena studied rather than describing it. We have therefore removed it from the sentence and added an additional sentence denoting their usefulness in understanding the phenomenon-line 301-305.